# HODA: Protecting DNNs Against Model Extraction Attacks via Hardness of Samples

## Abstract

Model Extraction attacks exploit the target model's prediction API to create a surrogate model in order to steal or reconnoiter the functionality of the target model in the black-box setting. Several recent studies have shown that a data-limited adversary who has no or limited access to the samples from the target model's training data distribution can use synthesis or semantically similar samples to conduct model extraction attacks. As the training process of DNN-based classifiers is done in several epochs, we can consider this process as a sequence of subclassifiers so that each subclassifier is created at the end of an epoch. We use the sequence of subclassifiers to calculate the hardness degree of samples. In this paper, we investigate the hardness degree of samples and demonstrate that the hardness degree histogram of a data-limited adversary's sample sequences is distinguishable from the hardness degree histogram of benign users' samples sequences, consisting of normal samples. Normal samples come from the target classifier's training data distribution. We propose Hardness-Oriented Detection Approach (HODA) to detect the sample sequences of model extraction attacks. The results demonstrate that HODA can detect the sample sequences of model extraction attacks with a high success rate by only monitoring 100 samples of them.

## 1 Introduction

Deep Neural Networks (DNNs) have shown impressive performance in various tasks in recent years that have encouraged the industry to deploy DNN-based models in a variety of real-world applications. Since the training process of DNNs and collecting training data is an expensive and tedious process, models are considered the intellectual property of organizations, and they must be kept secure. Therefore, models are often securely deployed on cloud servers, and only the creators can access the model parameters. Users are only allowed to query the model via a prediction API and receive predictions. Recent studies Tramèr et al. (2016); Papernot et al. (2017); Juuti et al. (2019); Orekondy et al. (2019); Jagielski et al. (2020) demonstrate that an adversary can exploit the prediction API of a target model to create a surrogate model in order to steal or reconnoiter the functionality of the target model. Such attacks are called *model extraction attacks*, and they violate the intellectual property of model owners. Furthermore, the surrogate model can be leveraged to conduct other attacks on the target model in black-box setting, such as adversarial example attacks Papernot et al. (2017); Juuti et al. (2019) and membership inference attacks Shokri et al. (2017).

Most of the model extraction attacks use the target model's prediction API to label an unlabeled dataset to create the surrogate model's training set. In most real-world settings, the adversary has no or limited access to samples from the target model's training data distribution, which is called *normal* or in-distribution samples. Hence, most proposed attacks in the previous studies use some form of Out-Of-Distribution (OOD) samples, such as synthesis Papernot et al. (2017); Juuti et al. (2019) or semantically similar samples to the target model's training set Orekondy et al. (2019); Pal et al. (2020) to conduct model extraction attacks. We focus on such attacks in this paper. There are two main approaches to defend against model extraction attacks, manipulating the target model outputs to prevent adversary from producing high-quality surrogate model Lee et al. (2019); Orekondy et al. (2020); Kariyappa & Qureshi (2020); Kariyappa et al. (2021b) and detecting the sample sequences of model extraction attacks Kesarwani et al. (2018); Juuti et al. (2019). Juuti et al. (2019) propose PRADA to detect samples sequence of model extraction attacks based on the distance among successive samples. We propose Hardness-Oriented Detection Approach (HODA), a new approach to

detect sample sequences of model extraction attacks, which outperforms PRADA by a large margin and has significantly less computational overhead.

Generally, the training process of DNN-based classifiers is done in several epochs, and the resulted classifier at the end of the last epoch is considered the final classifier. We can consider the training process of DNN-based classifiers as a sequence of subclassifiers in which the $i^{th}$ subclassifier is created at the end of the $i^{th}$ epoch. HODA uses a subset of subclassifiers to compute the hardness degree of samples. The hardness degree of a sample is equal to the index of the subclassifier that all subsequent subclassifiers agree with its predicted label for that sample. It is important to note that we must save subclassifiers in the training phase of a target classifier in order to use their predictions to calculate the hardness degree of new samples. We demonstrate that the hardness degree histogram of normal sample sequences is distinguishable from the hardness degree histogram of model extraction attack sample sequences, and HODA uses this property to detect sample sequences of model extraction attacks. For each user, HODA calculates the distance between the hardness degree histograms of the user's samples and normal samples, and if the distance is greater than a threshold, the user is detected as an adversary. HODA can detect JBDA Papernot et al. (2017), JBRAND Juuti et al. (2019), and Knockoff Net Orekondy et al. (2019) attacks with a high success rate by only monitoring 100 samples of attack. We demonstrate that HODA is also highly effective when the target classifier is trained using transfer learning.

**Contributions**. (i) We demonstrate that the hardness degree of a sample for a classifier pertains to the training data distribution of that classifier. (ii) We indicate that the hardness degree histogram of normal samples is distinct from the hardness degree histograms of model extraction attack samples. (iii) We propose HODA to detect the sample sequences of model extraction attacks.

## 2 RELATED WORK

**Model Extraction Attacks**: Primary model extraction attacks try to extract the exact value of parameters Lowd & Meek (2005); Tramèr et al. (2016) and hyperparameters Wang & Gong (2018) of shallow models. In recent years, the proposed attacks mainly aimed to steal or reconnoiter the functionality of deep neural networks by querying them in the black-box setting. It is often sensibly assumed in the literature that the adversary has no or limited access to samples from the training set distribution of target classifier. In order to overcome this issue, attacks generally use some form of out-of-distribution samples, such as synthesis or semantically similar samples, to create the surrogate classifier's training set. Knockoff Net Orekondy et al. (2019), ActiveThief Pal et al. (2020), and Copycat CNN da Silva et al. (2018) use semantically similar datasets to the target model's training set to train a surrogate classifier. In another line of studies, Papernot et al. (2017), Juuti et al. (2019), Yu et al. (2020), Truong et al. (2021), Kariyappa et al. (2021a), and Barbalau et al. (2020) use synthetic data to create the surrogate classifier's training set.

**Model Extraction Defenses**: Existing defense methods against model extraction attacks generally distribute into two branches: perturbation-based and detection-based. Perturbation-based defenses Lee et al. (2019); Orekondy et al. (2020); Kariyappa & Qureshi (2020) attempt to prevent adversaries from producing high-quality surrogate classifiers by adding perturbation to the target classifier outputs. Recently, Kariyappa et al. (2021b) proposed a new defense with the same goal as perturbation-based defenses, which does not perturb the target classifier outputs. Their approach employs an ensemble of diverse models to produce discontinuous predictions for out-of-distribution samples. Detection-based defenses attempt to detect the occurrence of model extraction attacks by observing successive input queries to the target classifier. Kesarwani et al. (2018) present a method to detect extraction attacks against Decision Tree models. PRADA Juuti et al. (2019) is the first proposed detection-based defense for DNN models. We propose a new defense detecting the sample sequences of model extraction attacks via hardness of samples.

Atli et al. (2020) demonstrate that several OOD detection approaches, such as Baseline Hendrycks & Gimpel (2017) and ODIN Liang et al. (2018), have poor performance in detecting Knockoff Net attack samples. Hence, they propose a new OOD detection approach that leverages a classifier to detect OOD samples. However, their approach only rejects OOD samples, and it does not have any detection mechanism to detect adversaries. Besides, the OOD detector is trained on samples from the same distribution used by the adversary to conduct Knockoff Net attacks, which is an unrealistic assumption in practice. Concurrent with our work, Zhang et al. (2021) and Pal et al. (2021) propose

SEAT and VarDetect to detect sample sequences of model extraction attacks, respectively. SEAT aims to detect model extraction attacks that use several similar samples to extract a target model, such as jacobian-based attacks (Papernot et al. (2017); Juuti et al. (2019)). Hence, SEAT is ineffective when an adversary uses natural samples that are not similar to each other, such as Knockoff Net attack. VarDetect uses Variational Autoencoders (VAs) and Maximum Mean Discrepancy (MMD) to detect model extraction attacks. VarDetect has only been evaluated on low-dimensional datasets. Regarding that VarDetect uses VAs and MMD, it is unclear how well it performs on high-dimensional datasets. Besides, it uses the ImageNet dataset to extract target classifiers trained on very structurally different datasets, such as F-MNIST and SVHN. HODA is evaluated on harder attacks using attack datasets that are structurally similar to the target classifier's dataset, such as using ImageNet dataset to extract Caltech256 target classifier. HODA can detect both jacobian-based and Knockoff Net attacks, and it performs well on high-dimensional datasets, such as Caltech256 and CUB200. Furthermore, unlike other work Atli et al. (2020); Kariyappa & Qureshi (2020); Kariyappa et al. (2021b), HODA only needs access to in-distribution samples to detect model extraction attacks.

## 3  MODEL EXTRACTION ATTACKS

The model extraction attack is one of the most serious threats against machine learning-based classifiers on remote servers, such as Machine Learning as a Service (MLaaS). The adversary's goal is to create a surrogate classifier $f_s$ that imitates a target classifier $f_t$ on task $T$. Most model extraction attacks exploit target model $f_t$ to label unlabeled samples to create the surrogate model's training set. The adversary sends sample $x_i$ to the target model and receives its output $f_t(x_i)$, and then she uses pair $(x_i, f_t(x_i))$ to train surrogate classifier $f_s$. The output type of target model can be label, label confidence, top-k values in probability vector, or the entire probability vector. We only consider label $\bar{f}_t(x_i)$ and the entire probability vector $f_t(x_i)$ as the output type of target classifiers in our experiments. There are two primary intents for adversaries to conduct model extraction attacks, *stealing* and *reconnaissance*.

**Stealing**: Producing a high performance classifier is an expensive and time-consuming process and requires computational resources and experts. Besides, given that DNNs need a large number of training samples, collecting data and labeling them is a complex and costly procedure for most real-world applications. Therefore, adversaries are motivated to take advantage of a target classifier to reduce the cost of creating a new classifier. The adversary's goal in stealing is to maximize the accuracy of surrogate model on data distribution $\mathcal{D}_T$. Hence, the adversary's goal is:

$$\text{Maximize} \quad P_{(x,y) \sim \mathcal{D}_T} \bar{f}_s(x) = y \tag{1}$$

**Reconnaissance**: The model extraction attacks can be used to conduct other attacks in the black-box setting, such as adversarial example attacks Papernot et al. (2017); Goodfellow et al. (2015) and membership inference attacks Shokri et al. (2017). The adversary's goal in reconnaissance is to maximize the *fidelity* among surrogate and target classifiers in order to increase the success rate of black-box attacks. Similar to Jagielski et al. (2020), we consider label agreement among surrogate and target classifiers as the fidelity metric on data distribution $\mathcal{D}_T$. Hence, the adversary's goal is:

$$\text{Maximize} \quad P_{(x,y) \sim \mathcal{D}_T} \bar{f}_s(x) = \bar{f}_t(x) \tag{2}$$

Proposed model Extraction attacks create the surrogate classifier training set $\mathbb{X}_s = \{(x_i, f_t(x_i))\}_{i=1}^{B}$ by various methods, where $B$ is the attack budget. The attack budget determines the number of samples that an adversary is allowed to send to the target classifier and receive their associated predictions. After creating $\mathbb{X}_s$, the adversary trains surrogate classifier $f_s$ to minimize empirical loss on $\mathbb{X}_s$. We suppose that the adversary knows the architecture and hyperparameters of the target classifier and uses them to train the surrogate classifier. It is noteworthy that our proposed defense is independent of surrogate classifiers' training process.

## 4  OUR PROPOSAL: HARDNESS-ORIENTED DETECTION APPROACH

### 4.1  HARDNESS DEGREE OF SAMPLES

The training process of a DNN-based classifier can be considered a sequence of subclassifiers called $F_{subclf}$ so that each subclassifier is created at the end of an epoch. Suppose that classifier $f_t$ is

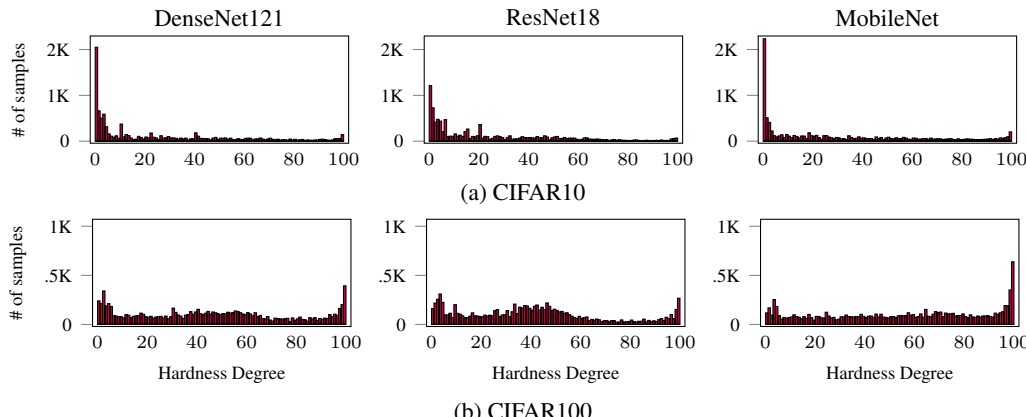

Figure 1: The hardness degree histograms of CIFAR10 and CIFAR100 test samples for DenseNet121, ResNet18, and MobileNet classifiers.

trained for $m$ epochs. The training process of classifier $f_t$ can be represented as the following sequence of subclassifiers:

$$F_{subclf} = < f_t^0, f_t^1, f_t^2, ..., f_t^{m-1} > \tag{3}$$

where subclassifier $f_t^i$ is created at the end of the $i^{th}$ epoch. We say sample $x_i$ is learned in epoch $e$ when $f_t^e$ is the first subclassifier that its predicted label is equal to all subsequent subclassifiers' predicted labels. Generally, as the number of epochs is increased, the performance of classifier $f_t$ is improved so that easier samples are learned in the early epochs, and harder ones are learned in the last epochs. Therefore, the *hardness degree* of sample $x_i$ for classifier $f_t$, which is displayed by $\phi_{f_t}(x_i)$, directly relates to the epoch number that $x_i$ is learned by $f_t$. Hardness degree of sample $x_i$ for classifier $f_t$ is defined as follows:

$$\phi_{f_t}(x_i) = e \quad \text{s.t.} \quad \forall j \in [e, m-1], \ \bar{f}_t^e(x_i) = \bar{f}_t^j(x_i), \ \bar{f}_t^e(x_i) \neq \bar{f}_t^{e-1}(x_i). \tag{4}$$

The hardness degree domain is dependent on the number of subclassifiers, and since we have $m$ subclassifiers, the hardness degree of a sample is in the range $[0, m-1]$. Since we want to calculate the hardness degree of samples in inference time, we need to save subclassifiers at the end of each or several epochs in the training phase of target classifiers to use them in the inference time. When a new sample arrives, it is fed to all loaded subclassifiers, and using their predictions, the hardness degree of that sample is computed. It is important to note that we do not use the true label of samples to calculate their hardness degree. As seen in equation (4), the hardness degree of samples is calculated only by the prediction of subclassifiers. It is noteworthy that we do not compute the hardness degree of the target classifiers' training samples throughout the paper and only compute the hardness degree of normal (test) or attack samples. Algorithm 1 in Appendix G describes computing the hardness of samples using $F_{subclf}$ in the inference time.

Table 1: The accuracy of classifiers on CIFAR10 and CIFAR100 test sets.

|  | Acc(%) | | |
| --- | --- | --- | --- |
|  | ResNet18 | DenseNet121 | MobileNet |
| CIFAR10 | 94.36 | 94.92 | 93.59 |
| CIFAR100 | 76.38 | 77.57 | 73.47 |

We train three various types of classifiers, including DenseNet121 Huang et al. (2017), ResNet18 He et al. (2016), and MobileNet Sandler et al. (2018), on CIFAR10 and CIFAR100 training sets for 100 epochs (details of datasets in Appendix A). All classifiers are trained using stochastic gradient descent with momentum 0.9 and batch size 128. The learning rate is 0.1 and it is scheduled to be decreased in each epoch by a constant factor 0.955. The accuracy of classifiers is presented in Table 1. We save all 100 subclassifiers in the training phase of each classifier and use them to calculate the hardness degree of samples. Figure 1 shows the hardness degree histogram of CIFAR10 and CIFAR100 test samples for various classifiers. The figure demonstrates that a large fraction of CIFAR10 test samples are easy, and many samples are learned in the first few epochs. However, the learning of CIFAR100 test samples is distributed over various epochs, and the number of hard samples is more than CIFAR10. Figure 6 in Appendix B demonstrates a strong positive correlation between the hardness degree of samples and the misclassification rate. As ResNet18 architecture

achieves strong performance on both datasets at a reasonable computational cost, we use this architecture for target classifiers in the rest of the paper. We conduct various model extraction attacks on two CIFAR10 and CIFAR100 target classifiers in the next subsection to depict the hardness degree histogram of their samples.

## 4.2 MODEL EXTRACTION ATTACKS SETUP

In line with prior work (Orekondy et al. (2020); Kariyappa & Qureshi (2020); Kariyappa et al. (2021b)), we select JBDA Papernot et al. (2017), JBRAND Juuti et al. (2019), and Knockoff Net Orekondy et al. (2019) model extraction attacks to evaluate our defense method. These attacks broadly represent two main strategies (synthesis or semantically similar samples) to conduct model extraction attacks. Jacobian-Based Dataset Augmentation (JBDA) Papernot et al. (2017) and its improvement (JBRAND) Juuti et al. (2019) assume that the adversary has access to a limited number of samples from the target classifier's training data distribution called seed samples, and they aim to augment seed samples using adversarial examples to increase the *fidelity* of the surrogate classifier to the target classifier. Orekondy et al. (2019) propose Knockoff Net (K.Net) attack that uses large public datasets that is semantically similar to the target classifier dataset to increase the *accuracy* of the surrogate classifier. We consider two versions of K.Net attack, K.Net CIFARX, and K.Net TIN. K.Net CIFARX attack uses CIFAR100 training set to extract CIFAR10 target classifier and vice versa. K.Net TIN employs TinyImageNet training set to extract target classifiers. More details about attacks and their implementations are presented in Appendix D.

To evaluate the performance of model extraction attacks, we use two ResNet18 classifiers being trained on CIFAR10 and CIFAR100 training sets as the target classifiers and conduct all four attacks on them. The default value of the attack budget in our experiments is 50000 (B=50K). Table 2 shows the accuracy and the fidelity of surrogate classifiers created by various model extraction attacks on CIFAR10 and CIFAR100 test samples. The results demonstrate that K.Net attacks have significantly better performance than jacobian-based attacks (JBDA and JBRAND), and when the output of target classifiers is probability vector, the performance of attacks is considerably increased.

Table 2: The Accuracy (Acc) and the Fidelity (Fid) of surrogate classifiers being created by four various model extraction attacks on two target classifiers CIFAR10 and CIFAR100. The output type of target classifiers can be Label or Probability Vector (Prob. Vec.).

| $f_t$ | Metric | Output type | JBDA | JBRAND | K.Net CIFARX | K.Net TIN |
|---|---|---|---|---|---|---|
| CIFAR10 ResNet18 (Acc: 94.36%) | Acc(%) | Prob. Vec. | 41.00 | 43.33 | 79.86 | 78.86 |
| | | Label | 34.57 | 34.35 | 66.88 | 71.29 |
| | Fid(%) | Prob. Vec. | 41.16 | 43.63 | 81.36 | 80.18 |
| | | Label | 34.86 | 34.45 | 67.98 | 72.43 |
| CIFAR100 ResNet18 (Acc: 76.38%) | Acc(%) | Prob. Vec. | 16.44 | 18.78 | 51.09 | 60.36 |
| | | Label | 8.62 | 8.07 | 23.20 | 32.88 |
| | Fid(%) | Prob. Vec. | 16.90 | 19.13 | 54.59 | 64.90 |
| | | Label | 8.91 | 8.29 | 24.72 | 34.58 |

## 4.3 HARDNESS DEGREE OF MODEL EXTRACTION ATTACK SAMPLES

Figure 2 depicts the hardness degree histogram of 50000 samples generated by various attacks for CIFAR10 and CIFAR100 target classifiers. In this experiment, the architecture of target classifiers is ResNet18. We also present the hardness degree histogram of attack samples when the architecture of target classifiers is Densenet121 in Appendix E. Figure 2 demonstrates that the samples generated by various attacks have a very small number of easy samples, and most samples have medium and high hardness degrees. However, Figure 1 indicates that a high number of normal samples that are from the same distribution as the target classifier's training set are easy.

To investigate more on the hardness degree of attack and normal samples, Figure 3 displays two-dimensional visualization of CIFAR10 test samples using t-SNE. Figure 3a uses the logits of the CIFAR10 classifier to visualize CIFAR10 test samples, and the color of each sample is determined by its label. This figure has ten sample clusters where most samples of each cluster are from one class. Figure 3b illustrates the hardness degree of CIFAR10 test samples for CIFAR10 target classifier and demonstrates that most of the easy samples are in the high-density regions inside clusters, and most of the hard samples are in the low-density regions at the borders of clusters. Figure 3c is similar to Figure 3b, but the hardness degree of each sample is calculated via CIFAR100 target classifier.

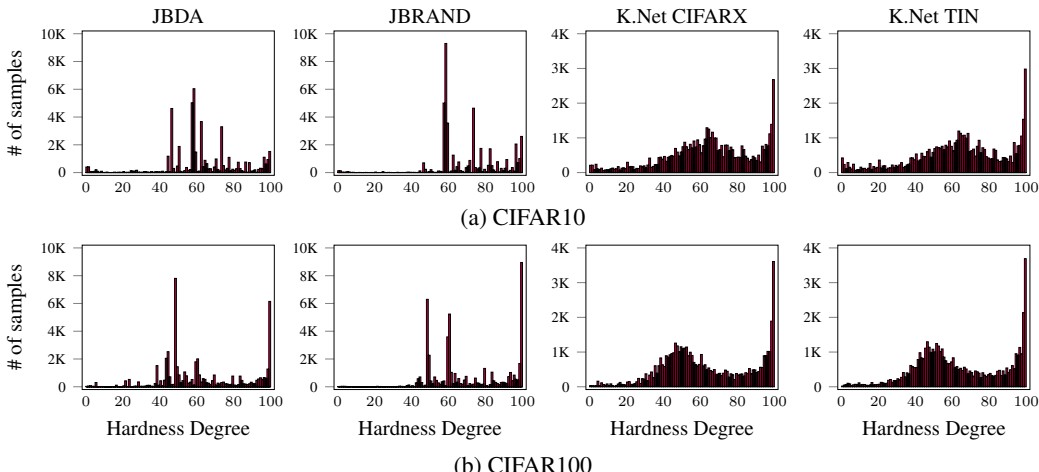

(a) CIFAR10

(b) CIFAR100

Figure 2: The hardness degree histograms of samples of four various model extraction attacks for CIFAR10 and CIFAR100 target classifiers. The budget of model extraction attacks is 50000.

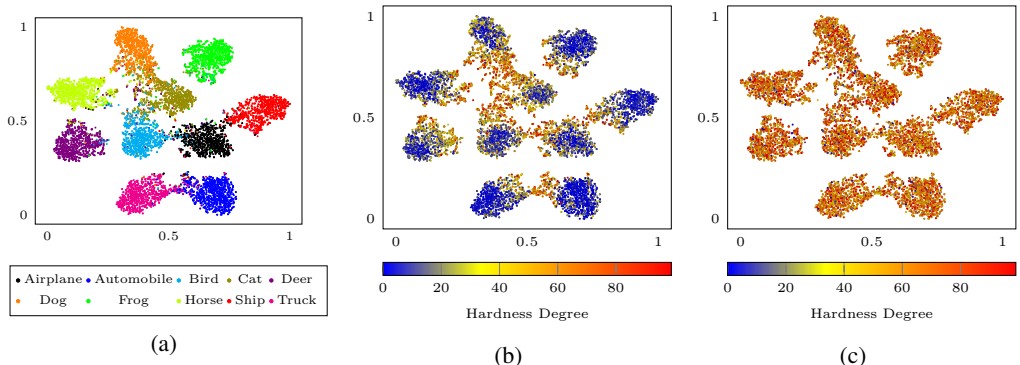

Figure 3: (a) Visualization of CIFAR10 test samples. (b) Hardness of CIFAR10 test samples for CIFAR10 classifier. (c) Hardness of CIFAR10 test samples for CIFAR100 classifier.

This figure demonstrates when the training data distribution of the classifier being used to calculate the hardness degree of samples becomes different from the distribution of CIFAR10 test samples, the hardness degree of a high number of samples is changed. Figure 3c shows hard and medium samples are distributed among clusters, and the number of easy samples is very small. Similar to Figure 3, we visualize CIFAR100 test samples and their hardness for CIFAR10 and CIFAR100 target classifiers in Appendix F. Based on our experiments, we consider hardness degree as an estimator of the empirical distribution of the target classifier's training data called $\mathcal{P}_{data}$. Easy samples are from the high probability region, and hard samples are from the low probability region of the input space. For example, if we suppose that $\mathcal{P}_{data}$ is a Gaussian-like distribution, easy samples are closer to the center of the distribution, and hard samples are in the tail of the distribution. Figures 3 and 8 show that easy and hard normal samples lie in the high- and low-density region of the input space. Figures 2 and 4 demonstrate that the number of easy samples among model extraction attack samples (OOD samples) is very small, which means that attack samples are from the low probability region of the input space. However, Figures 1 and 4 show a high number of normal samples are easy, which means that they are from the high probability region of the input space. We use histogram rather than hardness degree histogram in the rest of the paper for simplicity.

### 4.4 Hardness-Oriented Detection Approach

We propose Hardness-Oriented Detection Approach (HODA) to detect sample sequences of model extraction attacks. HODA requires normal histogram $H_n$ representing the histogram of normal

samples. When a new sample $x_i$ from user $u$ arrives, HODA calculates its hardness degree $\phi_{f_t}(x_i)$, and the histogram belongs to that user $H_u$ is updated. After the number of samples sent by user $u$ reaches a specific number $num_s$, HODA calculates Pearson distance between the histograms of normal samples $H_n$ and user samples $H_u$, and if the distance is greater than a threshold $\delta$, the user $u$ is detected as an adversary. Pearson Distance (PD) between two random variable $X$ and $Y$ is defined as follows:

$$PD(X,Y) = 1 - \frac{\text{Cov}(X,Y)}{\rho_X \rho_Y} \tag{5}$$

where $\text{Cov}(X,Y)$ is the covariance between random variables $X$ and $Y$, and $\rho_X$ is the standard deviation of random variable $X$. The output of Pearson distance is in the range [0,2]. To calculate the Pearson distance between two histograms, HODA first transforms histograms into probability vectors by dividing the value of histogram bins by the total number of samples in the histogram ($H_n/sum(H_n)$ and $H_u/sum(H_u)$) and then calculates the Pearson distance between them.

HODA uses normal sample set $S_{HODA}$ to create $H_n$ and calculate $\delta$. It randomly selects $num_{seq}$ sample sequences with size $num_s$ from the sample set $S_{HODA}$ and for each sample sequence, produces a histogram and adds it to the histogram set $HistSet$. The normal histogram $H_n$ is the average of all histograms in $HistSet$, and $\delta$ is the maximum Pearson distance between $H_n$ and all histograms in $HistSet$. Since $\delta$ is independent of attacks and only relies on normal samples, HODA is not dependent on any attacks. Notably, HODA does not need to save samples of each user or their hardness degrees. It only keeps a vector ($H_u$) that indicates the values of histogram bins for each user. Algorithm 2 in Appendix G describes HODA in details.

## 5   SETUP AND EVALUATION

Two normal sample sets $S_{HODA}$ and $S_u$ are required to evaluate the performance of HODA. $S_u$ is used to simulate benign users. We randomly select 40% and 60% of test samples of each dataset for $S_{HODA}$ and $S_u$, respectively. We randomly select $num_{seq} = 40000$ sequences with size $num_s$ from $S_{HODA}$ to create $H_n$ and calculate $\delta$. To evaluate the performance of HODA against model extraction attacks, we simulate 10000 benign users and 10000 adversaries for each attack. Each benign user sends a sequence of $num_s$ samples randomly selected from $S_u$, and each adversary sends a sequence of $num_s$ samples randomly selected from 50000 samples of attack in the order they were generated. So far, we have used 100 subclassifiers to calculate the hardness degree of samples. However, it may not be possible to classify each sample by a high number of subclassifiers in practice. So in order to reduce the computational cost of HODA, we use a subset of subclassifiers called $F_{subclf}$ to compute the hardness degree of samples. HODA only uses 11 subclassifiers to calculate the hardness degree of each sample, and these subclassifiers are saved in the training phase of target classifier $f_t$ at the end of each 10 epochs $F_{subclf} = <f_t^0, f_t^9, f_t^{19}, f_t^{29}, f_t^{39}, f_t^{49}, f_t^{59}, f_t^{69}, f_t^{79}, f_t^{89}, f_t^{99}>$. Algorithm 1 in Appendix G describes how hardness degree is computed using $F_{subclf}$ in details. Since the hardness degree domain depends on the number of subclassifiers, the hardness degree of a sample in HODA is in the range [0,10].

We compare the detection rate and the false-positive rate of HODA with PRADA. PRADA Juuti et al. (2019) declares that the histogram of minimum $L_2$ distance between a new sample and all previous samples of a benign user follows a Gaussian distribution. Hence, it uses the Shapiro-Wilk normality test to determine that a sample sequence belongs to a benign user or an adversary. Similar to HODA, PRADA also uses threshold $\delta$ to detect sample sequences of model extraction attacks, and $\delta$ is the only parameter of PRADA. Since PRADA needs to save each user's samples and calculate $L_2$ distance between them, it has a high computational overhead. Table 3

Table 3: The detection rate and False Positive Rate (FPR) of PRADA and HODA against four various model extraction attacks on CIFAR10 and CIFAR100 target classifiers.

| | | | | | Detection Rate of Attacks(%) | | | |
|---|---|---|---|---|---|---|---|---|
| | | $num_s$ | $\delta$ | FPR(%) | JBDA | JBRAND | K.Net CIFARX | K.Net TIN |
| CIFAR10 | PRADA | 100 | 0.818 | 0.01 | 0 | 0 | 0 | 0 |
| | | 500 | 0.973 | 0.05 | 96.7 | 94.2 | 4.4 | 1.6 |
| | HODA | 50 | 0.290 | 0.02 | 100 | 100 | 99.92 | 99.73 |
| | | 100 | 0.154 | 0.02 | 100 | 100 | 100 | 100 |
| CIFAR100 | PRADA | 500 | 0.550 | 0.01 | 0 | 0 | 0 | 0 |
| | | 1000 | 0.953 | 0.03 | 67.3 | 73.5 | 0 | 0 |
| | HODA | 50 | 0.716 | 0.02 | 94.65 | 100 | 90.68 | 89.06 |
| | | 100 | 0.349 | 0.02 | 100 | 100 | 100 | 100 |

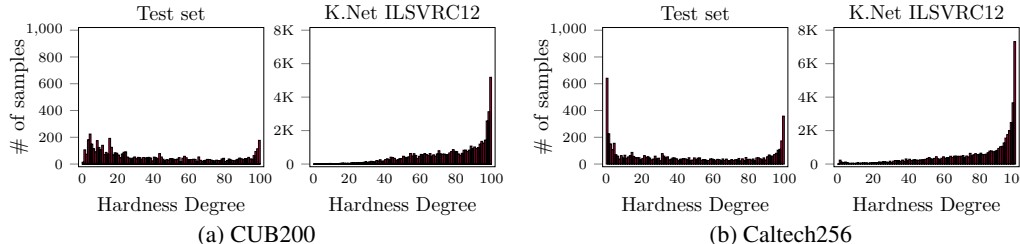

Figure 4: The left histogram in subfigures a and b shows the hardness degree histogram of CUB200 and Caltech256 test samples, respectively. The right histogram in each subfigure indicates the hardness degree histograms of K.Net ILSVRC12 attack samples on CUB200 (a) and Caltech256 (b) target classifiers.

indicates the detection rate and False Positive Rate (FPR) of PRADA and HODA against four various model extraction attacks on CIFAR10 and CIFAR100 target classifiers. We evaluate HODA when it only monitors 50 and 100 samples of each user ($num_s = 50$ and $num_s = 100$), and since PRADA needs to monitor more samples to detect attacks, we use larger $num_s$ to evaluate PRADA. PRADA and HODA have very low false-positive rates. False-Positive Rate (FPR) indicates the percentage of benign users' sample sequences wrongly detected as an attack. The results demonstrate that HODA is very effective against model extraction attacks, and it outperforms PRADA by a large margin. Since HODA does not rely on the distance between samples, it can detect knockoff Net attacks that use natural samples. HODA also has better performance on jacobian-based attacks. The runtime and the number of samples that need to be stored by PRADA depend on the attack. Nevertheless, for $num_s = 500$ and CIFAR10 target classifier, the average runtime of PRADA for each user is 0.47 seconds (prediction time not included) on Tesla K80 GPU, and 471 samples are stored for each user on average. For each user, the average runtime of HODA is 0.0012 seconds (prediction time not included), and it only stores a vector with size 11 representing a hardness degree histogram. Although HODA requires the predictions of 11 models to calculate the hardness degree of each sample, there is no sequential relationship between models, and they can predict in parallel, so HODA does not increase the prediction time of target models. Appendix I indicates the Pearson distance histogram of benign users and adversaries for all model extraction attacks. Appendix H introduces HODA-5 that uses five subclassifiers to calculate the hardness degree of samples. Table 6 shows the performance of HODA-5 against various model extraction attacks. Table 7 in Appendix L reports the accuracy of the K.Net attacks' surrogate classifiers for a defended adversary by HODA.

## 5.1 TRANSFER LEARNING

Transfer learning is a machine learning technique that initializes the parameters of the target task classifier using the parameters of a pre-trained source task classifier. We train two new target classifiers on CUB200 and Caltech256 datasets using transfer learning (details of datasets in Appendix A). The training process of new target classifiers is the same as CIFAR10 and CIFAR100 target classifiers (Section 4.1). We initialize the parameters of target classifiers from a pre-trained ImageNet Deng et al. (2009) classifier and train all layers of target clas-

Table 4: The detection rate and False Positive Rate (FPR) of HODA against K.Net ILSVRC12 attack.

| Target Model | $num_s$ | $\delta$ | FPR(%) | Detection Rate(%) K.Net ILSVRC12 |
|---|---|---|---|---|
| CUB200 | 50 | 0.973 | 0.01 | 97.50 |
| | 100 | 0.393 | 0.02 | 100 |
| Caltech256 | 50 | 0.694 | 0.01 | 99.98 |
| | 100 | 0.152 | 0.01 | 100 |

sifiers. Orekondy et al. (2020) indicate that jacobian-based model extraction attacks have very poor performance on high dimensional datasets. Thereby, we only evaluate the performance of target classifiers against K.Net ILSVRC12 attack. K.Net ILSVRC12 is the Knockoff Net attack that uses ILSVRC12 dataset as the surrogate classifier's training set. The budget of K.Net ILSVRC12 is 50000, and the output of target classifiers is the entire probability vector. The accuracy of CUB200 target classifier and its surrogate classifier is 73.7% and 59.3%, respectively, and the accuracy of Caltech256 target classifier and its surrogate classifier is 77.2% and 72.2%, respectively.

Figure 4 depicts the hardness degree histogram of CUB200 and Caltech256 test sets on the associated target classifier and also the hardness degree histogram of K.Net ILSVRC12 samples for both

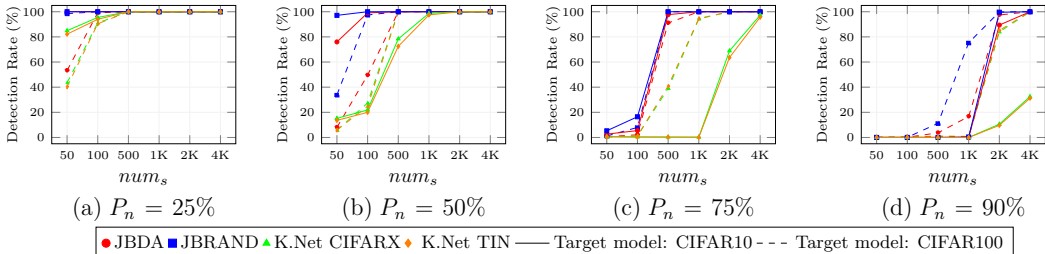

Figure 5: The detection rate of HODA for various percentages of normal samples $P_n$ over different values of $num_s$.

target classifiers. The figure demonstrates that the majority number of K.Net ILSVRC12 attack samples are hard (hardness degree > 70), and the number of easy samples (hardness degree < 30) is very small. We replicate the experiment of the previous section to evaluate the performance of HODA against K.Net ILSVRC12 attack with the same parameters. Table 4 shows the performance of HODA against K.Net ILSVRC12 attack on both target classifiers. The results demonstrate that even the starting point of target classifiers' parameters is not random, HODA is very effective in detecting K.Net ILSVRC12 attack.

## 6 DISCUSSION ON ADAPTIVE ADVERSARY

An adaptive adversary who is aware of HODA must send her queries based on the hardness degree histogram of normal samples to evade HODA. We consider two scenarios for an adaptive adversary to conduct model extraction attacks. In the first scenario, the adversary has no access to normal samples, and she only can use synthetic or semantically similar samples to extract the target model. There are two reasons why such attacks are hard to conduct. First, the adversary needs samples with various degrees of hardness; however, since the adversary has no access to the target classifier, she can not determine the hardness degree of her samples for the target classifier. Second, the adversary has no access to the histogram of normal samples to generate her samples based on it.

In the second scenario, we assume the adversary has access to a limited number of normal samples, and she can use normal samples to make her hardness degree histogram more similar to the hardness degree histogram of normal samples. To evaluate HODA in this scenario, we suppose that the adversary has access to 1000 normal samples from $S_{user}$ and she sends a sample sequence of which $P_n\%$ is filled by normal samples, and the rest is filled by model extraction attack samples. Notably, when the number of normal samples in the sequence exceeds 1000, the adversary sends duplicate normal samples. It is important to note that the cost of attack is increased by a factor of $\frac{1}{1-(P_n/100)}$ in this scenario. Figure 5 shows the detection rate of HODA for various $P_n$ over different values of $num_s$. The false-positive rate of all experiments is less than 0.2%. The figure demonstrates that increasing $num_s$ improves the detection rate of HODA. Except for K.Net attacks on CIFAR10 target classifier in $P_n = 90\%$, HODA can detect all attacks with a high success rate by increasing $num_s$. Due to the dataset limitation, we can not evaluate HODA for $num_s > 4000$. However, we think the detection rate of HODA against K.Net attacks on CIFAR10 target classifier in $P_n = 90\%$ will be improved for $num_s > 4000$. Altogether, we think the main challenge of an adaptive adversary to evade HODA is to collect easy samples, which are very rare in out-of-distribution samples based on our experiments.

## 7 CONCLUSIONS

This paper demonstrates that the hardness degree of samples is important in trustworthy machine learning. We investigated the hardness degree of samples and demonstrated that the hardness degree histogram of model extraction attack samples is different from the hardness degree histogram of normal samples. Using this observation, we proposed Hardness-Oriented Detection Approach (HODA) to detect sample sequences of model extraction attacks. HODA can detect the sample sequences of model extraction attacks with a high success rate by only monitoring 100 samples of attacks.

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

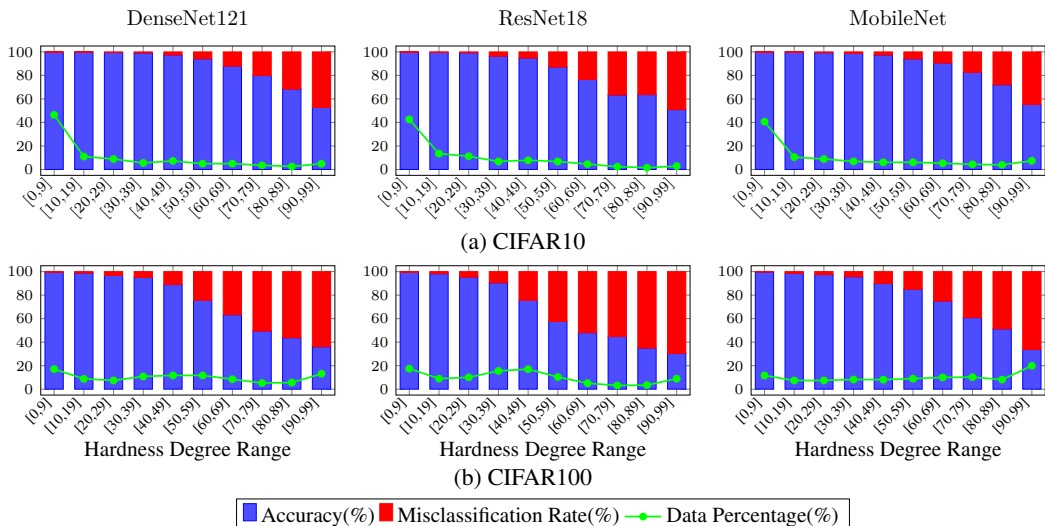

(a) CIFAR10

(b) CIFAR100

■ Accuracy(%) ■ Misclassification Rate(%) —●— Data Percentage(%)

Figure 6: Blue and red bars show the percentage of test samples in each range of hardness degrees, which are correctly or wrongly classified, respectively. For each range of hardness degrees, Data Percentage indicates the percentage of CIFAR10 and CIFAR100 test samples whose hardness degrees are in that range.

## A  DATASETS

**CIFAR10** Krizhevsky (2009): CIFAR-10 dataset consists of 60K $32 \times 32$ color images in 10 classes, including airplanes, cars, birds, cats, deer, dogs, frogs, horses, ships, and trucks. It has 6K images per class, where 5K images is in the training set and 1K images is in the test set.

**CIFAR100** Krizhevsky (2009): CIFAR100 dataset consists of 60K $32 \times 32$ color images in 100 classes. It has 600 images per class, where 500 images is in the training set and 100 images is in the test set.

**TinyImageNet** Le & Yang (2015): TinyImageNet is a subset of ILSVRC12 Deng et al. (2009) dataset, and contains 200 image classes. It has 500 training samples and 50 test samples for each class. The size of images is $64 \times 64$. We resize all images to $32 \times 32$.

**CUB200** Wah et al. (2011): CUB200 dataset contain 200 classes of bird categories. It consists of about 6K training and about 6K test samples. The size of images is $224 \times 224$.

**Caltech256** Griffin et al. (2007): Caltech256 dataset contain 256 classes of common objects categories. It consists of about 24K training and about 6K test samples. The size of images is $224 \times 224$.

**ILSVRC12** Deng et al. (2009): ILSVRC12 uses a subset of ImageNet and consists of 1.2 million training images, 50,000 validation images, and 100,000 test images. The dataset has 1000 classes and the size of images is $224 \times 224$.

**STL10** Coates et al. (2011): STL10 dataset consists of 13K $96 \times 96$ color images in 10 classes, including airplanes, cars, birds, cats, deer, dogs, monkeys, horses, ships, and trucks. It has 1.3K images per class, where 0.5K images is in the training set and 0.8K images is in the test set. We resize all images to $32 \times 32$.

## B  RELATIONSHIP BETWEEN THE ACCURACY OF CLASSIFIERS AND HARDNESS DEGREE OF SAMPLES

To assess the relationship between the hardness degree of samples and the misclassification rate, we compute the hardness degree of CIFAR10 and CIFAR100 test samples and then partition them into ten groups based on their hardness degree. It is important to note that the number of samples in each group is different. Afterward, we calculate what percentage of samples in each group is

classified incorrectly. Figure 6 demonstrates that the misclassification rate is increased by increasing the hardness degree of samples. In other words, there is a strong positive correlation between the hardness degree of samples and the misclassification rate. The figure indicates the percentage of samples in each hardness degree range by a green curve.

For example, the hardness degree of 40.65% of CIFAR10 test samples (4065 samples) is in the range [0,9] for MobileNet classifier, from which 99.88% is classified correctly, or the hardness degree of 7.4% of CIFAR10 test samples (740 samples) is in the range [90,99] for MobileNet classifier, from which 55.27% is classified correctly. More than 99% and 95% of samples being learned in the first 30 epochs (hardness degree $< 30$) are correctly classified in CIFAR10 and CIFAR100 test samples, respectively. On the other side, less than 55% and 36% of samples being learned in the last 10 epochs (hardness degree $\geq 90$) are correctly classified in CIFAR10 and CIFAR100 test samples, respectively.

## C   HARDNESS TRANSFERABILITY

In this section, we indicate that the hardness of samples is relatively transferable among various classifiers. We use three classifiers created in Section 4.1 and a new ResNet18 classifier in this experiment. Table 5 displays the Pearson correlation coefficients between hardness degree of CIFAR10 and CIFAR100 test samples for various pairs of classifiers. The results demonstrate a positive and strong correlation between the hardness degree of samples for various pairs of classifiers. Therefore, the hardness of samples is relatively transferable

Table 5: Pearson correlation coefficients between hardness degree of CIFAR10 and CIFAR100 test samples for various pairs of classifiers.

|  | Pearson Correlation Coefficient | |
|---|---|---|
|  | CIFAR10 | CIFAR100 |
| ResNet18-ResNet18 | 0.784 | 0.687 |
| ResNet18-DenseNet121 | 0.775 | 0.685 |
| ResNet18-MobileNet | 0.765 | 0.688 |
| DenseNet121-MobileNet | 0.769 | 0.706 |

between different classifiers. On the other side, it implies that the hardness degree of samples is relatively independent of the architecture of classifiers.

## D   DETAILS OF MODEL EXTRACTION ATTACKS

**Jacobian-Based Dataset Augmentation (JBDA)** Papernot et al. (2017): The goal of JBDA attack is to increase the fidelity of the surrogate classifier to the target classifier in order to produce adversarial examples for the target classifier in the black-box setting. The authors assume that the adversary has access to a limited number of normal samples called seed samples. JBDA augment seed samples using adversarial examples to improve the performance of surrogate model. The augmentation process is conducted in multiple rounds. In the first round, surrogate training set $\mathbb{X}_s$ is initialized by seed samples, and surrogate model $f_s$ is trained on $\mathbb{X}_s$. In the next rounds, sample set $S$ with size $\kappa$ is randomly selected from $\mathbb{X}_s$, and for each $x \in S$, adversarial example $x'$ is created using the following equation:

$$x' = x + \lambda.\text{sign}(J_{f_s}[f_t(x)]) \tag{6}$$

where $\lambda$ is step size and $J$ is the Jacobian function. Afterward, new adversarial examples are labeled by the target model, and they are added to $\mathbb{X}_s$. Lastly, surrogate model $f_s$ is trained on $\mathbb{X}_s$. The attack is implemented with $\lambda = 0.1$ and $\kappa = 2000$. The seed samples are selected from the test set of datasets. We use 500 (50 for each class) and 1000 (10 for each class) samples of CIFAR10 and CIFAR100 test sets for seed samples, respectively.

**Jacobian-Based Random Target (JBRAND)** Juuti et al. (2019): The goal of JBRAND is to improve the performance of JBDA. It perturbs each sample in multiple iterations to generate more powerful adversarial examples and generates targeted adversarial examples with random targets. We generate three adversarial examples with random targets for each sample and use the same seed samples as JBDA. Each sample is perturbed in five iterations with $\epsilon = \frac{64}{225 \times 5}$. The attack is implemented with $\lambda = \frac{64}{255}$ and $\kappa = 2000$.

**Knockoff Net (K.Net)** Orekondy et al. (2019): Knockoff Net attack uses large public datasets that are semantically similar to the target model's training samples to create the surrogate model's train-

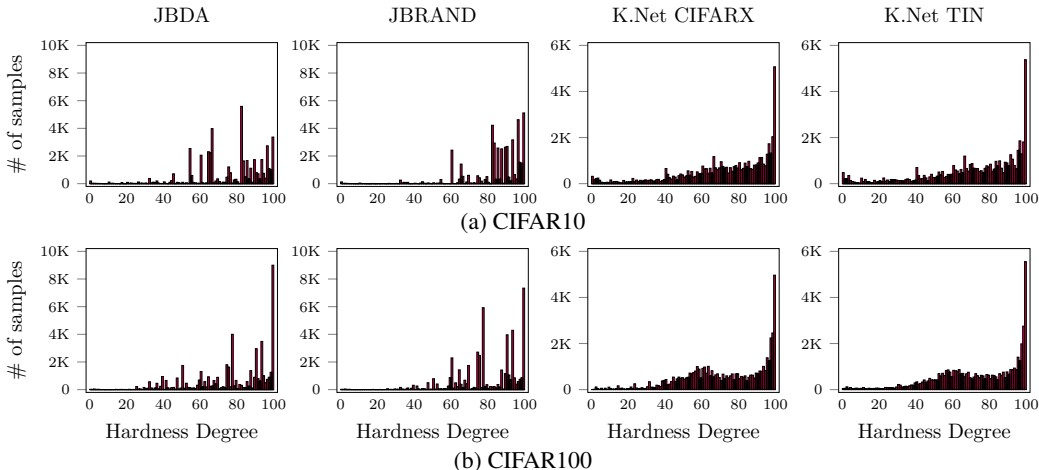

(a) CIFAR10

(b) CIFAR100

Figure 7: The hardness degree histograms of samples of four various model extraction attacks on CIFAR10 and CIFAR100 target classifiers. The budget of model extraction attacks is 50000. The architecture of target classifiers is DenseNet121.

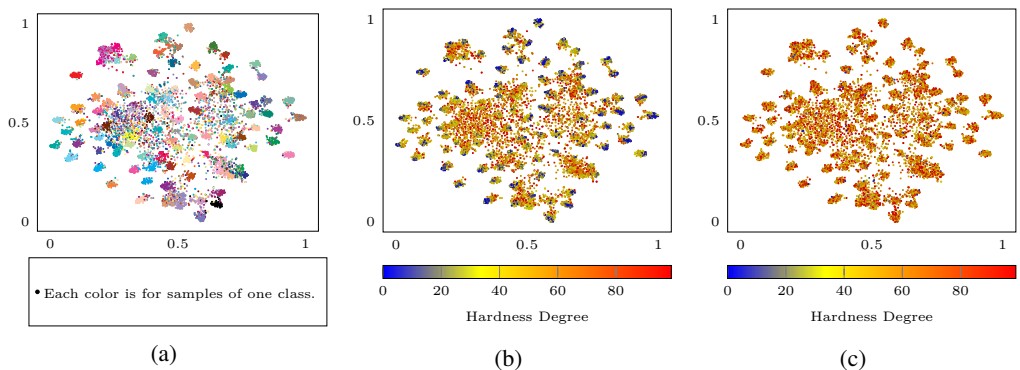

Figure 8: (a) Visualization of CIFAR100 test samples. (b) Hardness of CIFAR100 test samples for CIFAR100 classifier. (c) Hardness of CIFAR100 test samples for CIFAR10 classifier.

ing set. It has adaptive and random strategies to select the surrogate classifier's training set, which both use semantically similar samples. Since the adaptive strategy has very marginal benefits, we only consider the random strategy to implement this attack. K. Net randomly selects a subset of a public dataset and labels them using the target classifier to create $\mathbb{X}_s$. Finally, it uses $\mathbb{X}_s$ to train surrogate classifier $f_s$.

## E  DENSENET121 TARGET CLASSIFIERS

Figure 7 shows the hardness degree histograms of samples of four model extraction attacks on CIFAR10 and CIFAR100 target classifiers. The architecture of target classifiers is DenseNet121. As seen in Figure 7, the hardness degree histogram of model extraction attack samples is distinguishable from the hardness degree of normal samples (Figure 1) for DenseNet121 target classifiers.

## F  VISUALIZATION OF CIFAR100 SAMPLES AND THEIR HARDNESS

Figure 8 displays a two-dimensional visualization of CIFAR100 test samples using t-SNE. Figure 8a uses the logits of CIFAR100 classifier to visualize CIFAR100 test samples, and the color of each sample is determined by its label. Figures 8b and 8c show the hardness degree of CIFAR100 test samples for CIFAR100 and CIFAR10 target classifiers, respectively.

# G  HODA ALGORITHM

---

**Algorithm 1** Hardness Degree Computation

---

**Inputs**: $x$ is a sample and $F_{subclf}$ is a sequence of subclassifiers
**Outputs**: $degree$ is the hardness degree of sample $x$

1: **function** GETHARDNESSDEGREE($x, F_{subclf}$)
2:     $label \leftarrow$ None
3:     **for** $i \leftarrow 0, len(F_{subclf})$ **do**
4:         $pred\_vector \leftarrow F_{subclf}[i](x)$     // $F_{subclf}[i]$ is the $i^{th}$ subclassifier in sequence $F_{subclf}$
5:         $pred\_label \leftarrow$ argmax($pred\_vector$)
6:         **if** $pred\_label \neq label$ **then**
7:             $degree = i$
8:             $label = pred\_label$
9:     **return** $degree$
10: **end function**

---

**Algorithm 2** Hardness-Oriented Detection Approach (HODA)

---

**Inputs**: $S_{HODA}$ is a set of normal samples, $num_s$ is the size of sample sequences, $num_{seq}$ is the number of sample sequences, $F_{subclf}$ is the subclassifier sequence of target model, $NewQuery$ is the newest query being received by the target model, and $UserID$ is the identifier of owner of $NewQuery$.
**Outputs**: $H_n$ is the histogram of normal samples, $\delta$ is the attack detection threshold, $AttackAlarm$ declares the occurrence of attack.

1: **function** PEARSONDIST($H_n, H_u$)
2:     **return** $PD(H_n/Sum(H_n), H_u/Sum(H_u))$
3: **end function**
4: **function** HODAINITIALIZATION($S_{HODA}, num_s, num_{seq}, F_{subclf}$)
5:     $HistSet \leftarrow \emptyset$
6:     **for** $i \leftarrow 0, num_{seq}$ **do**
7:         $seq \leftarrow$ Randomly select $num_s$ samples from $S_{HODA}$
8:         $Hist \leftarrow \emptyset$
9:         **for** $s$ in $seq$ **do**
10:             $HD =$ GetHardnessDegree($s, F_{subclf}$)
11:             $Hist[HD] + = 1$
12:         $HistSet \leftarrow HistSet \cup Hist$
13:     $H_n \leftarrow Avg(HistSet)$
14:     $DistList \leftarrow \emptyset$
15:     **for** $Hist$ in $HistSet$ **do**
16:         $DistList$.append( PEARSONDIST($H_n, Hist$))
17:     $\delta \leftarrow Max(DistList)$
18:     **return** $H_n, \delta$
19: **end function**
20: **function** HODA($NewQuery, UserID, H_n, \delta, num_s, F_{subclf}$)
21:     $AttackAlarm \leftarrow False$
22:     $H_u \leftarrow$ GetUserHisogram($UserID$)
23:     $HD \leftarrow$ GetHardnessDegree($NewQuery, F_{subclf}$)
24:     $H_u[HD] + = 1$
25:     **if** $Sum(H_u) == Num_s$ **then**
26:         **if** PEARSONDIST($H_n, H_u$) $> \delta$ **then**
27:             $AttackAlarm \leftarrow True$
28:     SaveUserHistogram($H_u, UserID$)
29:     **return** $AttackAlarm$
30: **end function**

---

# H  HODA-5 (FIVE SUBCLASSIFIERS)

In Section 5, HODA uses 11 subclassifiers to calculate the hardness degree of samples. This section introduces HODA-5, which uses five subclassifiers ($F_{subclf} = < f_t^{19}, f_t^{39}, f_t^{59}, f_t^{79}, f_t^{99} >$) to calculate the hardness degree of samples. Hence, the domain of hardness degree is in the range [0,4]. Table 6 shows the performance of HODA-5 against model extraction attacks. The table demonstrates that although HODA-5 needs to monitor more samples ($num_s = 200$) to reach the performance of HODA (11 subclassifiers) against K.Net attacks, it is still very effective against model extraction attacks.

Table 6: The detection rate and False Positive Rate (FPR) of HODA-5 against four various model extraction attacks on CIFAR10 and CIFAR100 target classifiers.

|  |  | $num_s$ | $\delta$ | FPR(%) | Detection Rate of Attacks(%) | | | |
|  |  |  |  |  | JBDA | JBRAND | K.Net CIFARX | K.Net TIN |
|---|---|---|---|---|---|---|---|---|
| CIFAR10 | HODA-5 | 50 | 0.120 | 0.01 | 100 | 100 | 78.87 | 78.54 |
|  |  | 100 | 0.044 | 0.01 | 100 | 100 | 93.82 | 93.86 |
|  |  | 200 | 0.018 | 0.02 | 100 | 100 | 99.27 | 99.15 |
| CIFAR100 | HODA-5 | 50 | 0.370 | 0.01 | 90.37 | 100 | 80.0 | 80.62 |
|  |  | 100 | 0.140 | 0.02 | 100 | 100 | 97.08 | 97.79 |
|  |  | 200 | 0.060 | 0.01 | 100 | 100 | 99.87 | 99.85 |

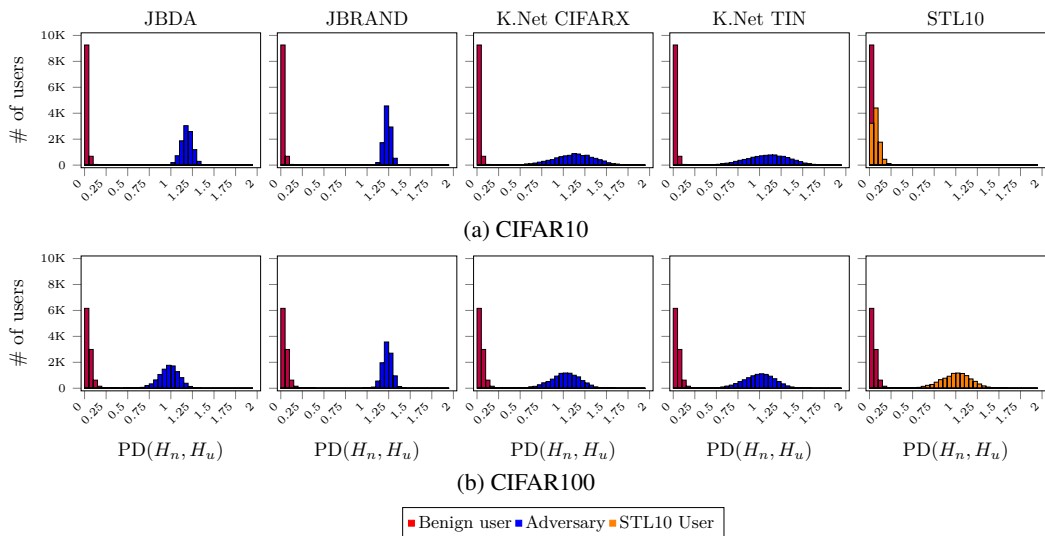

(a) CIFAR10

(b) CIFAR100

■ Benign user ■ Adversary ■ STL10 User

Figure 9: The histogram of Pearson distance between $H_n$ and 10000 benign users' hardness degree histogram and $H_n$ and 10000 adversaries' hardness degree histogram for various attacks. STL10 users are benign users for CIFAR10 target classifier and are adversaries for CIFAR100 target classifier.

## I   PEARSON DISTANCE HISTOGRAM

HODA creates a hardness degree histogram for each user, called $H_u$, and calculates Pearson distance between $H_u$ and normal hardness degree histogram $H_n$ (PD$(H_n, H_u)$). In section 5, we simulated 10000 benign users and 10000 adversaries for each attack. Figure 9 indicates the histogram of Pearson distance between $H_n$ and benign users' hardness degree histogram and also $H_n$ and adversaries' hardness degree histogram for $num_s = 100$. Notably, Pearson distance is in the range [0,2]. For CIFAR10 target classifier, Pearson distance between $H_n$ and hardness degree histogram of all 10000 benign users is less than 0.2, and as seen in Figure 9, Pearson distance between $H_n$ and hardness degree histogram of all 10000 adversaries of each attack is more than 0.2. For CIFAR100 target classifier, Pearson distance between $H_n$ and hardness degree histogram of all 10000 benign users is less than 0.35, and Pearson distance between $H_n$ and hardness degree histogram of all 10000 adversaries of each attack is more than 0.35. However, the confidence of HODA on CIFAR100 users is less than CIFAR10 users. We also consider 10000 STL10 users. STL10 users randomly select their samples from STL10 dataset (details in Appendix A). STL10 dataset has been inspired by the CIFAR10 dataset, and its images are obtained from the ImageNet dataset. It has the same classes as the CIFAR10 dataset, except instead of the frog class, it has a monkey class. We remove monkey images from STL10 dataset. Since classes of modified STL10 dataset are a subset of CIFAR10 classes, STL10 users use in-distribution samples for the CIFAR10 target classifier. However, similar to K.Net CIFARX attack on CIFAR100 target classifier, STL10 users use out-of-distribution samples for CIFAR100 target classifier. Since we suppose that only adversaries use out-of-distribution samples to extract a target model, STL10 users are adversaries for CIFAR100 target classifier and are benign users for CIFAR10 target classifier. The detection rate of HODA for STL10 users is 100% for CIFAR100 target classifier and 5.27% for CIFAR10 target classifier in $num_s = 100$. Hence, the false-positive rate of HODA for STL10 users, which are considered benign users, is 5.27% for CIFAR10 target classifier. By choosing a larger delta value (e.g., $\delta = 0.25$), the false-positive rate of HODA for STL10 users can be reduced to almost zero with almost no change in the detection rate of attacks.

## J   ADVERSARIAL EXAMPLES (AES)

Adversarial examples (AEs) are maliciously crafted inputs that cause the target classifier to misclassify them. There are numerous methods to generate adversarial examples such as L-BFGS Szegedy

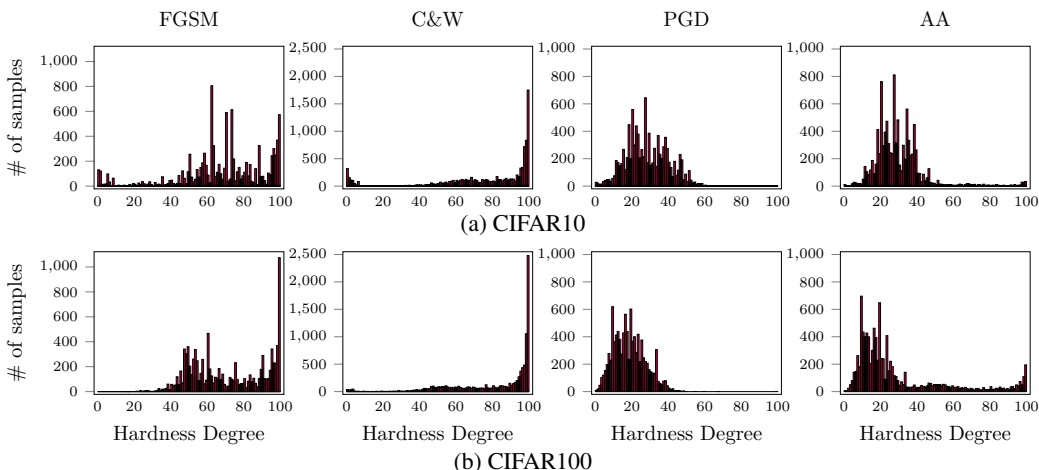

Figure 10: The hardness degree histograms of samples of four various adversarial example attacks on CIFAR10 and CIFAR100 target classifiers. Each attack uses 10000 natural samples in the test set associated with the target classifier dataset to create 10000 adversarial examples.

et al. (2014), FGSM Goodfellow et al. (2015), C&W Carlini & Wagner (2017), PGD Madry et al. (2018), and AutoAttack (AA) Croce & Hein (2020). To investigate the hardness of adversarial examples, we use FGSM ($\epsilon = 0.1$), C&W ($L_2$), PGD ($\epsilon = 8/255, \alpha = 3/255$), and AA ($L_\infty, \epsilon = 8/255$) attacks to generate adversarial examples on CIFAR10 and CIFAR100 test sets. The adversarial examples are created in the white-box setting, and they are untargeted. Figure 10 indicates the hardness degree of adversarial examples on CIFAR10 and CIFAR100 target classifiers. Although the distance between normal samples and adversarial examples is very small, Figure 10 demonstrates that the hardness degree histograms of adversarial examples are very different from normal samples (Figure 1). Most adversarial examples generated by FGSM and C&W are harder than adversarial examples generated by PGD and AA. We think this is because the size of perturbations added by FGSM and C&W is larger than PGD and AA. An intriguing observation is that almost all adversarial examples being generated by PGD are not hard (hardness degree < 70). AA has relatively the same behavior, and the number of its hard adversarial examples is very small.

## K    PERFORMANCE ANALYSIS OF MODEL EXTRACTION ATTACKS

To give new insight into model extraction attacks, we investigate the performance of model extraction attacks on normal samples with various levels of hardness. For this purpose, the test sets of CIFAR10 and CIFAR100 datasets are partitioned into 10 hardness groups based on hardness degree of samples. The hardness group $i$ consists of samples that their hardness degree is in range $[i \times 10, (i + 1) \times 10]$. Hence, the first hardness group consists of the easiest samples, and the last hardness group consists of the hardest ones. Figure 11 shows the accuracy and the fidelity of attacks over 10 hardness groups when the output of target classifier is the entire probability vector. The results demonstrate that the accuracy and the fidelity of all attacks are decreased as the hardness of samples is increased. We know from Figure 6 that the accuracy of target classifiers is decreased by increasing the hardness of samples. Figure 11 indicates the surrogate classifiers also have the same behavior.

The results demonstrate that the distance between the accuracy of target classifiers and surrogate classifiers (specially K.Net attacks) is increased by increasing the hardness of samples in the first hardness groups. However, the accuracy of the K.Net surrogate classifiers approaches the accuracy of target classifiers on the last two hardness groups. To investigate this observation, Figure 11 shows the percentage of samples being classified correctly by both surrogate classifier and target classifier for all attacks over various hardness groups with dashed lines. The results indicate that all samples correctly classified by surrogate classifiers are also correctly classified by target classifiers in the first two hardness groups. However, by increasing the hardness of samples, the surrogate classifiers correctly classify some samples that are not correctly classified by the target classifier,

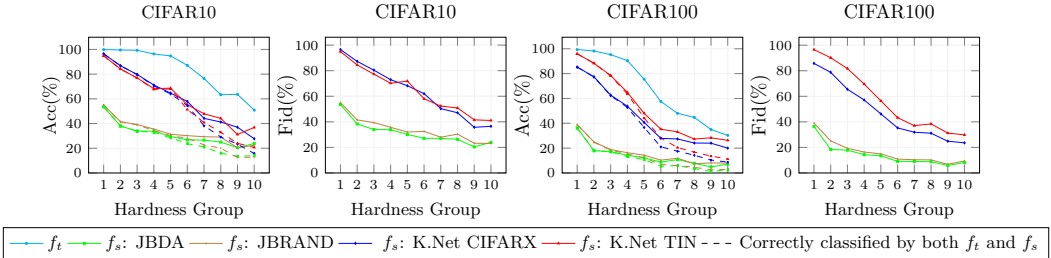

Figure 11: The accuracy and the fidelity of four surrogate classifiers over various hardness groups. The test set of each dataset is partitioned into 10 hardness groups so that hardness group 1 consists of the easiest samples and hardness group 10 consists of the hardest samples. The dashed lines indicate the percentage of samples being correctly classified by both target classifier $f_t$ and surrogate classifier $f_s$.

and the number of such samples is increased by increasing the hardness of samples. Jagielski et al. (2020) demonstrate that labels from the target classifier are more informative than dataset labels. We think the information in the labels that come from the target model causes the surrogate classifiers to correctly classify hard samples that are not correctly classified by the target classifier.

The fidelity of all surrogate classifiers is decreased by increasing the hardness of samples, which means that the disagreement among surrogate classifiers and target classifiers is raised on harder samples. An intriguing observation is that the fidelity of surrogate classifiers to the target classifiers on correctly classified samples by target classifiers is much more than wrongly classified samples.

## L    THE ACCURACY OF SURROGATE CLASSIFIERS OF DEFENDED ADVERSARIES

Table 3 shows that HODA detects all simulated adversaries by only monitoring 100 samples ($num_s = 100$) of each attack. Hence, adversaries can use the prediction of at most 100 samples to train the surrogate classifier. Table 7 reports the accuracy of K.Net surrogate classifiers on CIFAR10 and CIFAR100 test sets when the adversary only uses 100 samples of attack to train surrogate classifiers. Since simulated adversaries randomly select their samples from 50000 samples of each attack, we simulate ten adversaries for each experiment and report the average and standard deviation of the accuracy of ten trained surrogate classifiers for each attack. The training process and architecture of surrogate classifiers are similar to target classifiers, and the output of target classifiers is a probability vector.

Table 7: The average and standard deviation of the accuracy of surrogate classifiers for defended adversaries by HODA.

|  | Acc of Surrogate Classifier (%) | |
|---|---|---|
| $f_t$ | K.Net CIFARX | K.Net TIN |
| CIFAR10 (Acc: 94.36%) | $17.10 \pm 2.61$ | $17.77 \pm 1.76$ |
| CIFAR100 (Acc: 76.38%) | $2.71 \pm 0.38$ | $3.75 \pm 0.52$ |

## M    COMPREHENSIVE RELATED WORK

Some recent studies investigate the dynamic of DNNs training process. Hacohen et al. (2020); Frankle et al. (2020); Mangalam & Prabhu (2019) show that DNNs learn samples that are learnable by shallow models in early epochs of training before learning harder ones. Hacohen et al. (2020) demonstrate that DNNs learn samples in both training and test sets in a similar order. Pliushch et al. (2021) discuss correlation between learning order of samples with image statistics like segment count, edge strengths, image intensity entropy, and DCT coefficient matrix. Frankle et al. (2020) indicate that the DNN-based classifiers undergo substantial changes in the first few SGD iterations. Baldock et al. (2021) demonstrate a negative correlation between their measure of hardness (prediction depth) and learning events during training. They show that samples learned in later epochs have

higher prediction depth and confirm that neural networks learn easy samples first. In the following, we briefly review the most prominent model extraction attacks and defenses presented so far.

## M.1    MODEL EXTRACTION ATTACKS

For the first time, Lowd & Meek (2005) demonstrate the possibility of stealing simple linear machine learning models through only interaction with them. Tramèr et al. (2016) show the feasibility of model extraction attacks on commercial MLaaS. Papernot et al. (2017) and Juuti et al. (2019) investigate stealing DNN-based classifiers and propose jacobian-based model extraction attacks for creating a surrogate classifier in order to generate adversarial examples in the black-box setting. Chandrasekaran et al. (2020) explore the connection between active learning and model extraction attacks. They implement two query synthesis active learning algorithms to extract machine learning models, such as decision trees. Jagielski et al. (2020) use semi-supervised learning methods to improve the performance of model extraction attacks. Knockoff Net Orekondy et al. (2019), ActiveThief Pal et al. (2020), and Copycat CNN da Silva et al. (2018) use a semantically similar dataset to the target classifier's training set to create the surrogate classifier's training set. They employ different strategies for selecting samples from attack datasets to extract more information from the target classifier. Yu et al. (2020) employ active learning, transfer learning, and a new method for generating adversarial examples to improve model extraction attacks efficiency. A line of studies Truong et al. (2021); Kariyappa et al. (2021a); Barbalau et al. (2020) use synthetic data to create the training set of surrogate classifiers. Although their methods do not need to have access to natural samples, they send a high number of queries to the target classifier, which makes their methods impractical. For example, Truong et al. (2021) and Kariyappa et al. (2021a) send millions of queries to extract a CIFAR10 target classifier. While most model extraction attacks have focused on the vulnerabilities of image classifiers, recent studies demonstrate the vulnerability of NLP Krishna et al. (2020), Graph DNN He et al. (2021), and Reinforcement learning Chen et al. (2021) models against model extraction attacks. Another type of model extraction attack uses hardware side-channel vulnerabilities to extract a target classifier Zhu et al. (2021); Batina et al. (2019); Hong et al. (2018); Yan et al. (2020). However, these attacks have a very strong threat model and suppose the adversary has access to the hardware that hosts the target classifier.

## M.2    DEFENSES AGAINST MODEL EXTRACTION ATTACKS

Existing defense methods against model extraction attacks generally distribute into two branches: perturbation-based and detection-based defenses. Perturbation-based defenses Lee et al. (2019); Orekondy et al. (2020); Kariyappa & Qureshi (2020) attempt to prevent adversaries from producing high-quality surrogate classifiers by adding perturbation to the output of target classifier. These methods generate the perturbation with various strategies to minimize the accuracy of surrogate classifiers. Recently, Kariyappa et al. (2021b) proposed a new defense with the same goal as perturbation-based defenses, which does not perturb the output of target classifiers. Their approach employs an ensemble of diverse models to produce discontinuous predictions for out-of-distribution samples. Proposed detection-based defenses Kesarwani et al. (2018); Juuti et al. (2019) attempt to detect the occurrence of model extraction attacks by observing successive input queries to the target classifier. Kesarwani et al. (2018) propose a method to measure adversary perceived knowledge from target classifier, but this method only works for Decision Tree models. PRADA Juuti et al. (2019) is the first proposed detection-based defense for DNN models. PRADA uses the histogram of the minimum $L_2$ distance among a new sample and all previous samples to detect model extraction attacks. Aside from its high computational overhead, it has been shown that PRADA is unable to detect model extraction attacks when an adversary uses natural samples Pal et al. (2020). Watermarking neural networks Jia et al. (2021); Zhang et al. (2018); Szyller et al. (2021); Adi et al. (2018) is another type of defense against model extraction attacks. These methods prove ownership of a surrogate classifier instead of preventing the occurrence of model extraction attacks.

Atli et al. (2020) demonstrate that several OOD detection approaches, such as Baseline Hendrycks & Gimpel (2017) and ODIN Liang et al. (2018), have poor performance in detecting Knockoff Net attack samples. Hence, they propose a new OOD detection approach that leverages a classifier to detect OOD samples. However, their approach only rejects OOD samples, and it does not have any detection mechanism to detect adversaries. Besides, the OOD detector is trained on samples from the same distribution used by the adversary to conduct Knockoff Net attacks, which is an unreal-

istic assumption in practice. Concurrent with our work, Zhang et al. (2021) and Pal et al. (2021) propose SEAT and VarDetect to detect sample sequences of model extraction attacks, respectively. SEAT aims to detect model extraction attacks that use several similar samples to extract a target model, such as jacobian-based attacks (Papernot et al. (2017); Juuti et al. (2019)). Hence, SEAT is ineffective when an adversary uses natural samples that are not similar to each other, such as Knock-off Net attack. VarDetect uses Variational Autoencoders (VAs) and Maximum Mean Discrepancy (MMD) to detect model extraction attacks. VarDetect has only been evaluated on low-dimensional datasets. Regarding that VarDetect uses VAs and MMD, it is unclear how well it performs on high-dimensional datasets. Besides, it uses the ImageNet dataset to extract target classifiers trained on very structurally different datasets, such as F-MNIST and SVHN.

