# OpenReview forum: "HODA: Protecting DNNs Against Model Extraction Attacks via Hardness of Samples"
_ICLR.cc/2022/Conference — ICLR 2022 Submitted_

### Official Review · Reviewer_Mihr · 2021-10-20

**Correctness:** 3
**Technical Novelty And Significance:** 3
**Empirical Novelty And Significance:** 2
**Recommendation:** 6
**Confidence:** 4

**Main Review:**

This paper demonstrates that the hardness degree of samples is important in trustworthy machine learning. The authors investigated the hardness degree of samples and demonstrated that the hardness degree histogram of model extraction attack samples is different from the hardness degree histogram of normal samples.

My comments are as follows.

First, in section 4.1, this paper states a definition of the hardness degree, which is essential in this paper. However, I didn't see any significant result from Figure 2. Also, the meaning of accuracy in the Y-axis is unclear. In the last paragraph of page 4, it is not lucid to me how the authors get the results based on this figure.

Second, in section 4.2, the authors introduce the model extraction attacks and evaluate them. Nevertheless, I am confused about how to get the attack budget and how to get these images. I think the authors should add more details about that.

Next, in section 4.3, the authors calculate the hardness degree of the samples generated by attacks. However, I am wondering for the data samples that have a large hardness degree, is it easy to distinguish them from the data samples not used for span queries. As mentioned in the next section, the Pearson Distance will be very large. So, I don't think we can use the datasets that contain plenty of such samples to measure the sequences.

Finally, in section 5, the authors proposed the experiment setup and evaluation. However, I would like to know more evaluation metrics in the table, not only FPR.

Minor: in the last paragraph of page 7, it should FPR not PFR.


**Summary Of The Paper:**

This paper proposes Hardness-Oriented Detection Approach (HODA) to detect the sample sequences of model extraction attacks.


**Summary Of The Review:**

In general, I think this paper has merit.

---

> ### Author Response · Authors · 2021-11-22
> **Response**
>
> We are grateful for your thoughtful comments and suggestions. We hope to address the concerns raised by the reviewer below. We attached a PDF file in supplementary material where the main changes in the manuscript are highlighted in blue.
>
> **Q1. However, I didn't see any significant result from Figure 2.**
>
> **Response**: We first want to mention that since Figure 2 is not directly relevant to the main purpose of the paper, we have moved it to Appendix B and changed the figure to avoid misunderstanding. As mentioned in the paper, calculating the hardness degree of samples does not require the true label of samples. Nevertheless, since we had access to the true labels of CIFAR10 and CIFAR100 test samples, we depict Figure 2 (Figure 6) to give more intuition about the hardness degree of samples. To depict this figure, we first calculate the hardness degree of CIFAR10 and CIFAR100 test samples and then partition them into ten groups based on their hardness degree. Notably, the number of samples in each group is different. Afterward, we calculate what percentage of samples in each group is classified incorrectly. The figure demonstrates that the misclassification rate is increased by increasing the hardness degree of samples. In other words, there is a strong positive correlation between the hardness degree of samples and the misclassification rate. Figure 6 indicates the percentage of samples in each hardness degree range by a green curve. We have added an example to Appendix B to clarify Figure 6.
>
> **Q2. Nevertheless, I am confused about how to get the attack budget and how to get these images.**
>
> **Response**: We follow the previous work [1,2,3] to select the attacks budget. They use 50000 samples for the attack budget value. Due to page limitations, we transferred the details of the attacks to Appendix D, in which we explain how proposed attacks collect the surrogate classifier's training samples.
>
> **Q3. However, I am wondering for the data samples that have a large hardness degree, is it easy to distinguish them from the data samples not used for span queries.**
>
> **Response**: We are not sure we understand the question of the respected reviewer. If the question becomes clearer, we can provide more appropriate responses. First, we want to emphasize that HODA detects a sample sequence of attacks, not a single attack sample. Figure 1 demonstrates that most of the CIFAR10 test samples are easy, and the hardness degree of CIFAR100 test samples spread across histogram, but it has a wider peak on lower hardness degrees (easy samples). Figure 2 indicates that most attack samples are hard, and the hardness degree histogram of attacks samples is distinct from the hardness degree histogram of normal (test) samples (Figure 1). To detect sample sequences of model extraction attacks, HODA creates a hardness histogram from a holdout set of normal (test) samples that represent the hardness degree histogram of normal samples called $H_n$. For user u, HODA first creates a hardness degree histogram for samples of that user called $H_u$ and then computes Pearson distance between $H_u$ and $H_n$. If the distance exceeds a threshold, user u is an adversary.
>
> **Q4. Finally, in section 5, the authors proposed the experiment setup and evaluation. However, I would like to know more evaluation metrics in the table, not only FPR.**
>
> **Response:** To address the concern of the respected reviewer, Figure 9 shows the Pearson distance histogram of 10000 adversaries and 10000 benign users for all attacks in Appendix I. Figure 9 indicates the histogram of Pearson distance between $H_n$ and benign users' hardness degree histogram (samples of negative class in confusion matrix) and also $H_n$ and adversaries' hardness degree histogram (samples of positive class in confusion matrix) for $num_s=100$. Since HODA can completely separate benign users from adversaries, the AUC score of HODA for all four attacks is 1.
>
> **Q5. Minor: in the last paragraph of page 7, it should FPR not PFR.**
>
> **Response**: We corrected it.
>
> **References:**
>
> [1] S. Kariyappa, et al.. Protecting dnns from theft using an ensemble of diverse models. In ICLR, 2021b.
>
> [2] S. Kariyappa and M. K. Qureshi. Defending against model stealing attacks with adaptive misinformation. In CVPR, 2020.
>
> [3] T. Orekondy, et al.. Prediction poisoning: Towards defenses against DNN model stealing attacks. In ICLR, 2020.

---

### Official Review · Reviewer_QDdd · 2021-11-01

**Correctness:** 3
**Technical Novelty And Significance:** 2
**Empirical Novelty And Significance:** 3
**Recommendation:** 3
**Confidence:** 4

**Main Review:**

**Strengths**

1\. Novelty and certain benefits
- While there has been some recent attention to defenses via manipulation, detecting queries targeted for model stealing has receive little attention. The paper explores the latter line and demonstrates empirical improvements.
- The defense also provides some benefits e.g., defender does not need attacker's model architecture or parameters, it can be combined with manipulation-based defense strategies.

2\. Experiments
- I appreciate the authors comparing against multiple recent attack strategies and also evaluating in a variety of settings.

3\. Well-written
- The paper is well-written and was easy to follow.

**Major Concerns**

1\. Constraints / Severe assumptions
- Although the results appear promising, I am concerned that these results are a product of possibly impractical assumptions and would appreciate the authors' clarification:
- (a) examples can be precisely associated with a user $u$: what if the user $u$ performs a sybil attack and spreads the examples among multiple different user accounts? (which seems easily achievable in MLaaS scenarios.)
- (b) to obtain hardness values of a example now requires 10-100$\times$ as many predictions (to classify over all $f^m$ classifiers)
- (c) distribution(benign set) = distribution (training set): but this seems highly unlikely as benign distributions can have a very specific dataset bias (e.g., objects with white-backgrounds, images under poor lighting). I would guess in this case the benign user would be considered adversaries?

2\. Hardness analysis and comparison to OOD
- A key contribution of the paper is the concept of 'hardness' that captures some statistics of a set of examples and contrasts it with a normal set consisting of the victim's training examples. While the results using hardness appear promising, I find missing some additional analysis or discussion justifying 'hardness'. After all, the defense task is reduced to comparing two distributions and this is achievable by a variety of techniques (e.g., FIDs, Wasserstein, MMD). While the authors show that hardness is a good measure, I find lacking *why* or some in-depth analysis motivating hardness.
- I appreciate the empirical study in Sec. 4.3. that shows hardness captures OODness of samples. This makes me wonder if there are connections to existing OOD detection approaches (e.g., ODIN ICLR '18) and if one could instead simply use OOD detection techniques (which has significantly progressed recently) to defend against model stealing attacks.

3\. Experiments / Evaluation
- OOD baselines: Building on top of my previous concern, I would have appreciated comparison of HODA with OOD detection approaches (e.g., ODIN) or similar OOD techniques used within model stealing literature (e.g., Adaptive misinformation, Kariyappa and Qureshi CVPR '19). These seem more competitive and fairer to compare HODA with. Moreover, this seem more appealing from a defense perspective, since they typically perform example-level classification (vs. user-level) and as a result only certain examples can be rejected.
- Table 3-4: To claim that  HODA outperforms PRADA in Table 3-4, I find overall results missing e.g., FPR vs. detection rate as a result of modifying $\delta$, AUROC numbers. Going by the table alone, I am unsure if there exists a range of thresholds $\delta$ where HODA performs poorly.
- Simulating benign users: How are benign users $S_u$ simulated -- by using a held-out fraction of the victim's test set? If yes, this seems concerning as it is possible to design the approach or tune the hyperparameters and overfit to biases in the victim's distribution. One idea would be to construct the benign sample set by using images from CIFAR10 classes, but sampled from TinyImage or another dataset.


**Minor Concerns**

1\. Large hardness values -> noisy?
- For histograms in Fig. 1,3 wouldn't the values at higher hardness values be extremely noisy? For instance for a hardness value of 98 (at m=100 epochs), this can be achieved using only predicts at a single reference epoch (e=99). This seems to be case going by the curves where there is a big peak at higher hardness values. As a result, it might make sense to train the model instead of 200 epochs and estimate hardness values until 100.

**Summary Of The Paper:**

- The paper proposes a defense 'HODA' against DNN model stealing attacks.
- They key idea of the approach is a strategy that discriminates a set of 'benign' examples (i.e., similar to victim's data distribution) vs. 'surrogate' examples (i.e., from attacker's input distribution).
- The strategy involves mapping an example $x_i$ (benign/adversarial) to 'hardness' value which roughly indicates at which point in the training process the intermediate model converges on prediction of $x_i$. HODA exploits a property that the hardness distribution statistics varies between benign and surrogate sets.
- Experiments on standard datasets {CIFAR10, CIFAR100, CUB200, ...} and attacks {knockoff, JBDA, ...} indicates that HODA is able to achieve reasonable detection rates and FPR against attacks.

**Summary Of The Review:**

- The proposed approach 'HODA' extends the state of defenses against model stealing attacks by exploiting an interesting property of how an attacker's distribution contrasts a benign distribution. The experimental results indicate that indeed this property can be exploited to some extent to reject malicious queries.
- However, I am primarily concerned about (a) certain assumptions (being able to associate each example with the same user, distribution comparison) and (b) how this compares to simply performing OOD detection. In particular with (b), if the key idea to exploit that attacker's distribution is semantically different from the victim's, I would have appreciate a better comparison with OOD detection, or present some analysis/insights of the proposed 'hardness' justifying the choice.

---

> ### Author Response · Authors · 2021-11-22
> **Response 1/2**
>
> We are grateful for your thoughtful comments and suggestions. We hope to address the concerns raised by the reviewer below. We attached a PDF file in supplementary material where the main changes in the manuscript are highlighted in blue.
>
> **Q1. What if the user u performs a sybil attack and spreads the examples among multiple different user accounts?**
>
> **Response**: It is a valid concern, but it has to be considered that the budget of most model extraction attacks is in the order of 50000 to 100000 queries (millions in some cases). HODA detects model extraction attacks by observing only 50-100 queries, which makes the Sybil attack scenario highly difficult in the real world. Besides, other detection methods [1,2,3] are also vulnerable to Sybil attacks.
>
> **Q2. To obtain hardness values of a example now requires 10-100 ×as many predictions**
>
> **Response**: Using an ensemble of classifiers is a prevalent technique in ML literature, and many papers have used this technique to improve their methods. For example, [4] loads five classifiers in parallel. HODA uses 11 subclassifiers which are several checkpoints in the training process of a target classifier, and they have no independent training process. We show that 11 models are sufficient to detect model extraction attacks with a high success rate. In Appendix H, we introduce HODA-5 that uses five subclassifiers to calculate the hardness degree of samples. Table 6 shows the performance of HODA-5 against various model extraction attacks. The table demonstrates that HODA-5 is very effective against model extraction attacks. Finally, if the defender can provide resources so that subclassifiers predict in parallel, HODA has a very low computational cost (memory and runtime) compared to its competitors. Moreover, HODA-5 is almost as good as HODA and has a lower computational cost.
>
> **Q3. distribution(benign set) = distribution (training set)**
>
> **Response**: Except for one work [2], all previous work [4,5,6] that the performance of their approaches dependent on the OODness of samples uses a holdout set of test samples to simulate benign users to the best of our knowledge. Pal et al. [2] have considered a holdout set of classes that are not used during training (called AltPD) to simulate benign users (we think a user that uses samples from different classes than the classes of the target classifier is not a benign user!). However, they used the target classifier dataset to create the holdout set. Hence, they also use the same dataset of the target classifier to simulate benign users.
>
> To address the concern of the respected reviewer, we evaluate HODA against 10000 STL10 users and report the results of this experiment in Appendix I. STL10 users randomly select their samples from STL10 dataset, which has almost the same classes as CIFAR10. Hence, they are benign users for CIFAR10 target classifier. For $num_s=100$, the false-positive rate of HODA for STL10 users is 5.27%. As seen in Figure 9, by choosing a larger delta value (e.g., $\delta=0.25$), the false-positive rate of HODA for STL10 users can be reduced to almost zero with almost no change in the detection rate of attacks. Also, we think that by increasing $num_s$, the FPR of HODA against STL10 users is decreased. Overall, we think the performance of HODA against unseen STL10 users is decent, and HODA has the potential to adapt to various environments.
>
> **Q4. I find lacking why or some in-depth analysis motivating hardness.**
>
> **Response**: We add a paragraph to Section 4.3 (last paragraph) to justify the concept of hardness degree.
>
> "Based on our experiments, we consider hardness degree as an estimator of the empirical distribution of the target classifier's training data called $P_{data}$. Easy samples are from the high probability region, and hard samples are from the low probability region of the input space. For example, if we suppose that $P_{data}$ is a Gaussian-like distribution, easy samples are closer to the center of the distribution, and hard samples are in the tail of the distribution. Figures 3 and 8 show that easy and hard normal samples lie in the high- and low-density region of the input space. Figures 2 and 4 demonstrate that the number of easy samples among model extraction attack samples (OOD samples) is very small, which means that attack samples are from the low probability region of the input space. However, Figures 1 and 4 show a high number of normal samples are easy, which means that they are from the high probability region of the input space."
>
>
> **Q5. This makes me wonder if there are connections to existing OOD detection approaches.**
>
> **Response**: Study [5] (introduced by other reviewers) has done the requested experiment. Atli et al. [5] demonstrate that the detection rate of ODIN [7] and Baseline [8] for Knockoff Net attack samples is low (Table 4 in [5]). Hence, they propose a new OOD detection approach that leverages a classifier to detect OOD samples. ...

---

> > ### Author Response · Authors · 2021-11-22
> > **Response 2/2**
> >
> > However, their approach [5] only rejects OOD samples, and it does not have any detection mechanism to detect adversaries. Moreover, the OOD detector is trained on samples from the same distribution used by the adversary to conduct Knockoff Net attacks, which is an unrealistic assumption in practice.
> >
> > **Q6. OOD baselines: Building on top of my previous concern, I would have appreciated comparison of HODA with OOD detection approaches (e.g., ODIN).**
> >
> > **Response**: As mentioned in response to the previous question, OOD detection approaches, such as ODIN [7], have low performance in detecting attack samples. As mentioned by the respected reviewer, Adaptive Misinformation (AM) [6] has used an OOD detector. AM assumes that the defender has access to an OOD dataset to train the OOD detector, which is a strong assumption. HODA needs access only to normal (in-distribution) samples to detect model extraction attacks. Moreover, since AM is not confident about the OOD detector's predictions, it uses the OOD detector in a fuzzy manner and does not explicitly use it to detect OOD samples. AM uses the OOD detector's prediction to adjust the size of perturbation added to the target classifier's output. However, HODA explicitly detects model extraction attacks with high confidence (Figure 9).
> >
> > We can compare the performance of HODA with AM in terms of the accuracy of a defended adversary's surrogate classifier. Since EDM [4] outperforms AM (Table 2 in [4]), we compare HODA with EDM instead of AM. EDM also uses an OOD dataset to train its models. To compare HODA with EDM, Table 7 in Appendix L reports the accuracy of the surrogate classifiers of a defended adversary by HODA. EDM conducts K.Net CIFARX attacks on both CIFAR10 and CIFAR100 target classifiers. EDM decreases the accuracy of K.Net CIFARX surrogate classifier from 85.39% to 68.50% (0.8x) for CIAFR10 target classifier and from 53.04% to 41.16% (0.77x) for CIFAR100 target classifier. HODA decreases the accuracy of K.Net CIFARX surrogate classifier from 79.86% to 17.10% (0.21x) for CIAFR10 target classifier and from 51.09% to 2.71% (0.05x) for CIFAR100 target classifier. Finally, except for concurrent work [2,3], we have compared HODA with the only detection-based approach in the literature (PRADA). The limitation of the concurrent work has been added to the Related Work section.
> >
> > **Q7. Table 3-4: Going by the table alone, I am unsure if there exists a range of thresholds δ where HODA performs poorly.**
> >
> > **Response**: To address the concern of the respected reviewer, Figure 9 in Appendix I shows the Pearson distance histogram of 10000 adversaries and 10000 benign users for all attacks. Figure 9 indicates the histogram of Pearson distance between $H_n$ and benign users' hardness degree histogram (samples of negative class in confusion matrix) and also $H_n$ and adversaries' hardness degree histogram (samples of positive class in confusion matrix) for $num_s=100$. Since HODA can completely separate benign users from adversaries, the AUC score of HODA for all four attacks is 1.
> >
> > **Q8. Simulating benign users**
> >
> > **Response**: The same response as Q3.
> >
> > **Q9. For histograms in Fig. 1,3 wouldn't the values at higher hardness values be extremely noisy?**
> >
> > **Response**: The respected reviewer is right. The range of hardness degree is dependent on the number of subclassifiers. Given our computational resources, we have trained target classifiers in 100 epochs. If the target classifiers train in more epochs (for example, 200), the peak at the end of histograms disappears, and histograms smoothly go down. Nevertheless, since HODA only uses eleven subclassifiers to detect model extraction attacks, increasing the number of target classifier epochs to more than 100 does not impact the performance of HODA.
> >
> > References:
> >
> > [1] M. Juuti, et al.. PRADA: protecting against DNN model stealing attacks. In IEEE European S&P, 2019.
> >
> > [2] S. Pal, et al.. Stateful detection of model extraction attacks. CoRR, abs/2107.05166, 2021. URL https://arxiv.org/abs/2107.05166.
> >
> > [3] Z. Zhang, et al.. Seat: Similarity encoder by adversarial training for detecting model extraction attack queries. In AISec, 2021.
> >
> > [4] S. Kariyappa, et al.. Protecting dnns from theft using an ensemble of diverse models. In ICLR, 2021b.
> >
> > [5] G. Atli, et al.. Extraction of complex dnn models: Real threat or boogeyman? In Engineering Dependable and Secure Machine Learning Systems, 2020.
> >
> > [6] S. Kariyappa and M. K. Qureshi. Defending against model stealing attacks with adaptive misinformation. In 2020 CVPR, 2020.
> >
> > [7] S. Liang, Y. Li, and R. Srikant. Enhancing the reliability of out-of-distribution image detection in neural networks. In ICLR, 2018.
> >
> > [8] Hendrycks and K. Gimpel. A baseline for detecting misclassified and out-of-distribution examples in neural networks. In ICLR, 2017.

---

> ### Author Response · Authors · 2021-12-04
> **A Gentle Reminder**
>
> Dear reviewer QDdd,
>
> We hope that you've had a chance to read our response and the revised paper. We would really appreciate a reply before the end of the discussion period as to whether we have addressed your concerns or if any additional concerns remain. We are happy to address any remaining concerns and are eagerly looking for your feedback on the revised paper.
>
> Thanks so much!

---

### Official Review · Reviewer_UQxC · 2021-11-02

**Correctness:** 1
**Technical Novelty And Significance:** 2
**Empirical Novelty And Significance:** 2
**Recommendation:** 1
**Confidence:** 4

**Main Review:**

The authors use the ordered ensemble of checkpoints to give a numerical hardness value to image samples and argue that the distribution of benign test samples by hardness differs from attack samples. Using this, distance is computed between holdout set histogram and test time histogram. If it exceeds a threshold, an attack is detected.

The definition of hardness appears to have conceptual flaws:
* It does not account for mis-classified samples. A mis-classified sample $x$ should intuitively be one of the most hard samples and should be assigned the highest hardness degree $m-1$ (where $m-1$ is the last epoch). Among the epoch-wise checkpoints, $x$ could get a wrong label $l$ in some epoch $e < m-1$ and the model could continue to label it as $l$ in the later epochs. So the hardness degree $\phi_{f_t}(x) = e < m-1$, which is counter-intuitive.
* Let $A$ be a model that is not well-trained (e.g., trained using a very small learning rate) and the original labels of the training samples do not change at all. Then all the samples will get hardness degree of $0$! In comparison, a well-trained model $B$ will have samples falling into higher values of hardness. Thus, by the hardness definition in the paper, the well-trained model $B$ would find the same set of samples "harder" than an ill-trained model. There does not seem to be any correlation between model accuracy on a dataset and hardness of the dataset.
* There is no guarantee in general that model training is monotonic for each example, that is worst to better. So concluding hardness based on the latest epoch number at which its label stabilizes is not indicative of its hardness.

Because of the conceptual flaws, I am not convinced about validity of the proposed approach.

There are several limitations of the experiments:
* The paper compares only against PRADA, which is not designed to work for attacks based on natural samples (e.g., Knockoff nets used in the paper). There are at least two defenses which are designed to detect such attacks: Atli et al. "Extraction of Complex DNN Models: Real
Threat or Boogeyman?" and Pal et al. "Stateful detection of model extraction attacks". The authors should compare against these defenses. The paper incorrectly states that PRADA is the only defense in this space.
* To construct benign users, the authors take a split from the original test distribution. In practice, a benign user can provide samples that could come from similar distributions and not necessarily from the same distribution. The authors should conduct experiments to check false positives for such cases, e.g., MNIST vs color MNIST.
* The results on adaptive attacks show that HODA requires an order of magnitude more samples (50-100 to 500-1K) to detect a decent number of attacks. For the CIFAR10 model, it requires even more samples. HODA requires running each sample on multiple checkpoints, which can be expensive in the adaptive case due to increasing number of samples required for detection.
* Appendix H gives trend over different hardness degrees for the target and extracted models. This however does not tell us the model accuracy numbers exactly. The authors should report model accuracy for these models and for models extracted when HODA is applied.


**Summary Of The Paper:**

In this paper, the authors propose HODA (hardness oriented detection approach) to detect model extraction attacks against DNNs. They use the checkpoints during training of the model to define hardness of a sample. Hardness of a sample is the epoch from which the model assigns the same class to the sample. During inference, Pearson distance between the histograms of holdout set samples and test samples is used to detect an attack. HODA is evaluated on image datasets and models. It is shown to detect three classes of attacks and is analyzed against an adaptive attack.


**Summary Of The Review:**

The paper is targeting a challenging problem. However, the central concept of hardness of samples as defined in the paper appears flawed. The paper has several limitations when in comes to experimental evaluation.

---

> ### Author Response · Authors · 2021-11-22
> **Response 1/3**
>
> We are grateful for your thoughtful comments and suggestions. We hope to address the concerns raised by the reviewer below. We attached a PDF file in supplementary material where the main changes in the manuscript are highlighted in blue.
>
> Respectfully, based on the summary of the paper, we think the respected reviewer misinterpreted the definition of hardness. We try to clarify the concept of hardness degree in the following.
>
> 1- HODA only uses the prediction of subclassifiers (several checkpoints in the training phase of the target classifier) to calculate the hardness degree of samples. True label of samples has no role in calculating the hardness degree of samples. The hardness degree of samples is computed through the prediction of the subclassifiers in the inference time. The hardness degree of a sample is equal to the index of the subclassifier that all subsequence subclassifiers agree with its predicted label. In other words, the hardness degree of a sample is the index of the subclassifier that the target model does not change its decision about that sample in subsequent subclassifiers. We add algorithm 1 to Appendix G to clarify how HODA computes the hardness degree of samples using a sequence of subclassifiers.
>
> 2- In all experiments, measuring hardness happens in the inference time. There is no histogram for the hardness degree of target classifier’s training data. Figure 1 shows the hardness degree histogram of CIFAR10 and CIFAR100 test samples, and Figure 2 displays the hardness degree histogram of samples of various attacks.
>
> 3- We consider test samples as normal (in-distribution) samples that are from the target classifier's training data distribution. HODA uses a holdout set of test samples called $S_{HODA}$ to create the hardness degree histogram of normal samples called $H_n$. We suppose that benign users also send a sequence of normal samples that come from a different holdout set of test samples called $S_u$.
>
> 4- For each user, HODA calculates the hardness degree of that user's samples and creates a hardness degree histogram called $H_u$. The respected reviewer mentioned, "During inference, Pearson distance between the histograms of training set samples and test samples is used to detect an attack.", which is not correct. Pearson distance is calculated between the user's hardness degree histogram $H_u$ and hardness degree histogram of normal samples $H_n$ (both in inference time without access to true label of samples). If the distance exceeds a threshold, the user is an adversary. Therefore, HODA does not use the hardness degree histogram of target classifier’s training set samples. Generally, HODA compares the hardness degree histogram of a holdout set of test samples and the hardness degree histogram of attack samples in order to detect model extraction attacks.
>
> We have added a paragraph to Section 4.1 to clarify the concept of hardness degree.
>
> **Q1. It does not account for mis-classified samples.**
>
> **Response**: Since there is no access to the true label of samples in the inference time, we cannot determine the hardness degree of samples based on their true labels. However, since we have access to the true label of CIFAR10 and CIFAR100 samples, we assess the relationship between the hardness of samples and the misclassification rate in Figure 6. Figure 6 in Appendix B (Figure 2 in the previous version) demonstrates that there is a strong positive correlation between the hardness of samples and the misclassification rate. We add a paragraph to Section 4.3 to justify the concept of hardness degree.
>
> "Based on our experiments, we consider hardness degree as an estimator of the empirical distribution of the target classifier's training data called $P_{data}$. Easy samples are from the high probability region, and hard samples are from the low probability region of the input space. For example, if we suppose that $P_{data}$ is a Gaussian-like distribution, easy samples are closer to the center of the distribution, and hard samples are in the tail of the distribution. Figures 3 and 8 show that easy and hard normal samples lie in the high- and low-density region of the input space. Figures 2 and 4 demonstrate that the number of easy samples among model extraction attack samples (OOD samples) is very small, which means that attack samples are from the low probability region of the input space. However, Figures 1 and 4 show a high number of normal samples are easy, which means that they are from the high probability region of the input space."
>
> HODA uses this observation to detect model extraction attacks. We do not claim that our definition of hardness perfectly differentiates samples based on their true hardness, but given the experiments, we think it is a good method to estimate the hardness degree of samples.

---

> > ### Author Response · Authors · 2021-11-22
> > **Response 2/3**
> >
> > **Q2. Let A be a model that is not well-trained (e.g., trained using a very small learning rate) and the original labels of the training samples do not change at all.**
> >
> > **Response**: We suppose that the target model is a well-trained model. An ill-trained model reduces the quality of hardness estimation. The quality of the hardness estimation is dependent on the quality of the target classifier. As the quality of the target classifier increases, the quality of the hardness estimation increases too. Figure 6 (Figure 2 in the previous version) demonstrates that there is a strong positive correlation between the hardness of samples and the misclassification rate. This figure demonstrates that the misclassification rate is increased by increasing the hardness degree of samples.
> >
> > **Q3. There is no guarantee in general that model training is monotonic for each example, that is worst to better.**
> >
> > **Response**: Our definition of hardness degree has been inspired by the definition of learning order in [5]. The concept of learning order among samples of a (real) dataset is a known subject in the literature. Hacohen et al. demonstrate that DNN models learn samples of training and test sets in a similar order, and this pattern is even shared among DNN models with different architectures. An intriguing observation is that DNN models with higher capacity (e.g., more layers) learn almost all of the samples learned by lower capacity DNN models in similar order and then start to learn harder samples (same observation reported in [8]). In [6], the authors have investigated learning order and discussed it is almost independent of architecture, training setup, and rather more dependent upon the dataset itself. Parallel to our work, Baldock et al. [7] demonstrate a negative correlation between their measure of hardness (prediction depth) and learning events during training. They show that samples learned in later epochs have higher prediction depth and confirm that neural networks learn easy samples first. Our results in Figure 6 are consistent with the experiments of this paper. Most of the mentioned papers have defined the hardness of samples in the training phase of DNNs, but we focus on the hardness of samples in the inference time.
> >
> > **Q4. The paper compares only against PRADA, which is not designed to work for attacks based on natural samples (e.g., Knockoff nets used in the paper).**
> >
> > **Response**: [2] has been published on arXiv less than three months before the ICLR 2022 submission deadline. This paper is concurrent with our work, and we mention it in the related work. Unfortunately, we had not seen [3], and we sincerely apologize for this mistake. [1] also has not mentioned [3]. The proposed method in [3] is an out-of-distribution (OOD) detection method, and the authors do not propose any model extraction detection mechanism in their paper. They try to decrease the accuracy of the surrogate classifier by rejecting OOD samples. Therefore, in addition to the strong threat model of this paper about the knowledge of the defender, it is not clear how we compare our method with this paper. In another parallel work, Zhang et al. [4] have also introduced another model extraction detection-based method. However, we still believe that PRADA is the only defense that is comparable to HODA. [2] and [4] are parallel to our work, and we had not enough time to compare HODA with them. We have added a paragraph to the Related Work section of the manuscript to describe these three studies and their limitations.
> >
> > **Q5. To construct benign users, the authors take a split from the original test distribution.**
> >
> > **Response**: Except for one work [2], all previous work [1,3,9] that the performance of their approaches dependent on the OODness of samples uses a holdout set of test samples to simulate benign users to the best of our knowledge. Pal et al. [2] have considered a holdout set of classes that are not used during training (called AltPD) to simulate benign users (we think a user that uses samples from different classes than the classes of the target classifier is not a benign user!). However, they used the target classifier dataset to create the holdout set. Hence, they also use the same dataset of the target classifier to simulate benign users. ...

---

> > > ### Author Response · Authors · 2021-11-22
> > > **Response 3/3**
> > >
> > > To address the concern of the respected reviewer, we evaluate HODA against 10000 STL10 users and report the results of this experiment in Appendix I. STL10 users randomly select their samples from STL10 dataset. STL10 dataset has been inspired by the CIFAR10 dataset, and its images are obtained from the ImageNet dataset. It has the same classes as the CIFAR10 dataset, except instead of the frog class, it has a monkey class. We remove monkey images from STL10 dataset. Since classes of modified STL10 dataset are a subset of CIFAR10 classes, STL10 users use in-distribution samples for the CIFAR10 target classifier, and they are benign users. For $num_s=100$, the false-positive rate of HODA for STL10 users is 5.27%. As seen in Figure 9, by choosing a larger delta value (e.g., $\delta=0.25$), the false-positive rate of HODA for STL10 users can be reduced to almost zero with almost no change in the detection rate of attacks. Also, we think that by increasing $num_s$, the FPR of HODA against STL10 users is decreased. Overall, we think the performance of HODA against unseen STL10 samples is decent, and HODA has the potential to adapt to various environments. For example, delta can be selected by the sample sequence profile of users that are certainly benign.
> > >
> > > **Q6. The results on adaptive attacks show that HODA requires an order of magnitude more samples (50-100 to 500-1K) to detect a decent number of attacks.**
> > >
> > > **Response**: Using an ensemble of classifiers is a prevalent technique in ML literature, and many papers have used this technique to improve their methods. For example, [1] loads five classifiers in parallel. HODA uses 11 subclassifiers which are several checkpoints in the training process of a target classifier, and they have no independent training process. In Appendix H, we introduce HODA-5 that uses five subclassifiers to calculate the hardness degree of samples. Table 6 shows the performance of HODA-5 against various model extraction attacks. The table demonstrates that HODA-5 is very effective against model extraction attacks. It detects jacobian-based model extraction attacks by only monitoring 100 samples of each adversary. HODA-5 needs to monitor more samples ($num_s=200$) to reach the performance of HODA (11 subclassifiers) against K.Net attacks. Finally, if the defender can provide resources so that subclassifiers predict in parallel, HODA has a very low computational cost (memory and runtime) compared to its competitors. Moreover, HODA-5 is almost as good as HODA and has a lower computational cost.
> > >
> > > **Q7. Appendix H gives trend over different hardness degrees for the target and extracted models.**
> > >
> > > **Response**: The accuracy and fidelity of surrogate (extracted) classifiers have been reported in Table 2. Table 7 in Appendix L reports the accuracy of the K.Net attacks' surrogate classifiers for a defended adversary by HODA.
> > >
> > > **References**:
> > >
> > > [1] S. Kariyappa, et al.. Protecting dnns from theft using an ensemble of diverse models. In ICLR, 2021b.
> > >
> > > [2] S. Pal, et al.. Stateful detection of model extraction attacks. CoRR, abs/2107.05166, 2021. URL https://arxiv.org/abs/2107.05166.
> > >
> > > [3] G. Atli, et al. Extraction of complex dnn models: Real threat or boogeyman? In Engineering Dependable and Secure Machine Learning Systems, 2020.
> > >
> > > [4] Z. Zhang, et al.. Seat: Similarity encoder by adversarial training for detecting model extraction attack queries. In AISec, pp. 37–48, 2021.
> > >
> > > [5] G. Hacohen, et al.. Let’s agree to agree: Neural networks share classification order on real datasets. In ICML, 2020.
> > >
> > > [6] I. Pliushch, et al.. When deep classifiers agree: Analyzing correlations between learning order and image statistics. CoRR, abs/2105.08997, 2021. URL https://arxiv.org/abs/2105.08997.
> > >
> > > [7] R. Baldock, et al.. Deep learning through the lens of example difficulty. CoRR, abs/2106.09647, 2021. URL https://arxiv.org/abs/2106.09647.
> > >
> > > [8] K. Mangalam and V. Prabhu. Do deep neural networks learn shallow learnable examples first? In Workshop on Identifying and Understanding Deep Learning Phenomena at 36th ICML, 2019.
> > >
> > > [9] S. Kariyappa and M. K. Qureshi. Defending against model stealing attacks with adaptive misinformation. In CVPR, 2020.

---

> > > > ### Comment · Reviewer_UQxC · 2021-12-06
> > > > **Thank you**
> > > >
> > > > Thank you very much for your detailed response and the updated write-up. The additional experiments are helpful too. Thank you for pointing out that the term "training time" was inadvertently used in summarizing the approach. I have updated the review to address it.

---

> > > > > ### Author Response · Authors · 2021-12-09
> > > > > **Thank you**
> > > > >
> > > > > Dear reviewer UQxC,
> > > > >
> > > > > We truly appreciate your reply, and we are glad that our responses were helpful. Considering your very positive reply, we would really appreciate it if you could please consider raising your score.
> > > > >
> > > > > Thank you.

---

> ### Author Response · Authors · 2021-12-04
> **A Gentle Reminder**
>
> Dear reviewer UQxC,
>
> We hope that you've had a chance to read our response and the revised paper. We would really appreciate a reply before the end of the discussion period as to whether we have addressed your concerns or if any additional concerns remain. We are happy to address any remaining concerns and are eagerly looking for your feedback on the revised paper.
>
> Thanks so much!

---

### Official Review · Reviewer_qQMY · 2021-11-04

**Correctness:** 3
**Technical Novelty And Significance:** 3
**Empirical Novelty And Significance:** 3
**Recommendation:** 5
**Confidence:** 4

**Main Review:**

**Strong points:**
1. The method outperforms previous detection-based defense called PRADA.
2. The method is well-described but some more intuition why it works well would be helpful. Overall, the paper is well-written.

**Weak points:**
1. The storage overhead is substantial if the method has to store checkpoints after every epoch. However, it is shown that this requirement can be alleviated by storing checkpoints only every 10th epoch and it is sufficient to maintain a high detection rate as well as the low false-positive rate of the HODA defense. Additionally, defenses usually come at a cost, so this can be justified. The authors could compare to PRADA not only in terms of storing the hardness histograms but also add the size of the saved checkpoints.
2. The computational cost of the method is increased by a number X of saved checkpoints in comparison to the standard inference. Usually, the DNN models are large and if only a single K80 GPU is available, then HODA cannot be executed in parallel (not sufficient memory). Additionally, the GPU load + energy cost is increased by X as well.

**Other points:**
1. There are other detection-based defenses, e.g. [1] and [2].
2. Minor inconsistent naming convention, e.g.: CIFAR10 and Cifar10.
3. Why is MobileNet performing the best across the 3 models in Figure 2?
4. Many transfer learning methods freeze all but the last (classification) layer. How does HODA perform in this setting (or only if few last layers are retrained)? What is the dependence between how many last layers are re-trained and the detection rate of HODA?
5. In Appendix G, PGD examples are not hard with hardness degree **<** 70.

**References**

[1] Stateful Detection of Model Extraction Attacks. Soham Pal, Yash Gupta, Aditya Kanade, Shirish Shevade. https://arxiv.org/abs/2107.05166

[2] Extraction of Complex DNN Models: Real Threat or Boogeyman? Buse Gul Atli, Sebastian Szyller, Mika Juuti, Samuel Marchal, N. Asokan. https://arxiv.org/abs/1910.05429


**Summary Of The Paper:**

HODA is a new defense against model extraction attacks. It is a detection method that determines if a given user is adversarial based on the hardness of users' queries. For a given sample, the hardness metric is computed as the epoch number of a subclassifier after which the sample is classified correctly. The method generates histograms for in-distribution data and the histograms of each user's queries. The Person distance is computed between the histograms and if the distance is greater than a certain threshold $\delta$ then the user is classified as malicious. The results show that the hardness degree histogram is an indicator of dataset distribution. In general, the in-distribution samples have significantly lower hardness levels than samples used by adversaries (the adversarial samples are OOD natural or synthetic data).

**Summary Of The Review:**

The paper presents a new detection-based defense against model extraction attacks that performs better than the previous (directly comparable) defense called PRADA. However, the defense is expensive in terms of computation and storage usage.

---

> ### Author Response · Authors · 2021-11-22
> **Response 1/2**
>
> We are grateful for your thoughtful comments and suggestions. We hope to address the concerns raised by the reviewer below. We attached a PDF file in supplementary material where the main changes in the manuscript are highlighted in blue.
>
> Based on your summary of our paper, we want to mention that the hardness degree of samples is not the epoch number of a subclassifier after which the sample is classified correctly. Since we calculate the hardness degree of samples in inference time, we do not access the true label of samples. Hence, we cannot find which subclassifier correctly classifies a sample. The hardness degree of samples is computed through the prediction of subclassifiers. The hardness degree of a sample is equal to the index of the subclassifier that all subsequence subclassifiers agree with its predicted label. In other words, the hardness degree of a sample is the index of the subclassifier that the target model does not change its decision about that sample in subsequent subclassifiers. We have added a paragraph to Section 4.1 to clarify the concept of hardness degree.
>
> **Q1. The storage overhead is substantial if the method has to store checkpoints after every epoch.**
>
> **Response**: Using an ensemble of classifiers is a prevalent technique in ML literature, and many papers have used this technique to improve their methods. All papers that use an ensemble of classifiers need more storage. For example, [1] loads five classifiers in parallel. HODA uses 11 subclassifiers which are several checkpoints in the training process of a target classifier, and they have no independent training process. We show that 11 models are sufficient to detect model extraction attacks with a high success rate. In Appendix H, we introduce HODA-5 that uses five subclassifiers to calculate the hardness degree of samples (5 models are loaded in parallel). Table 6 shows the performance of HODA-5 against various model extraction attacks. It detects jacobian-based model extraction attacks by only monitoring 100 samples of each adversary. HODA-5 needs to monitor more samples ($num_s=200$) to reach the performance of HODA (11 subclassifiers) against K.Net attacks.
>
> The scalability of a defense similar to HODA is calculated with respect to the number of users. For example, PRADA stores almost 450 samples for each user (in $num_s=500$) to detect model extraction attacks (1000 users * 224$\times$224$\times$3 image size * 450 samples = 63.08 Gigabytes). HODA needs to save only a vector with size 11 representing a hardness degree histogram for each user (1000 user * 11 histogram bins = 10.7 Kilobytes). The storage cost of HODA for a worthy classifier that needs to be protected is reasonable. Besides, PRADA must calculate L2 distance between samples of each user. As mentioned in the paper, 11 models can predict in parallel, but if it is not possible, the prediction time of HODA will be 11 times more than PRADA. Accordingly, the size of the saved checkpoints of HODA is also 11 times more than PRADA (in HODA-5 is 5 times more). Finally, if the defender can provide resources so that subclassifiers predict in parallel, HODA has a very low computational cost (memory and runtime) compared to its competitors. HODA-5 is almost as good as HODA and has a lower storage cost.
>
> **Q2. The computational cost of the method is increased by a number X of saved checkpoints in comparison to the standard inference.**
>
> **Response**: The computational cost of HODA is increased by increasing the number of subclassifiers. However, we show that 11 models are sufficient to detect model extraction attacks with a high success rate, and we also demonstrate in Appendix H that HODA-5 (calculating hardness degree by five subclassifiers) is also effective in detecting model extraction attacks. We could not afford several GPUs, but providing several GPUs is feasible for many organizations in the real world. Since 11 models can predict in parallel, HODA does not impose any delay to prediction time. Hence, we consider the same prediction time for HODA and PRADA. The prediction time has not been included in the reported runtime of PRADA and HODA. We add to the paper that prediction time is not included in the reported runtime of HODA and PRADA.

---

> > ### Author Response · Authors · 2021-11-22
> > **Response 2/2**
> >
> > **Q3. There are other detection-based defenses, e.g. [1] and [2].**
> >
> > **Response**: [2] has been published on arXiv less than three months before the ICLR 2022 submission deadline. This paper is concurrent with our work, and we mention it in the related work. Unfortunately, we had not seen [3], and we sincerely apologize for this mistake. [1] also has not mentioned [3]. The proposed method in [3] is an out-of-distribution (OOD) detection method, and the authors do not propose any model extraction detection mechanism in their paper. They try to decrease the accuracy of the surrogate classifier by rejecting OOD samples. Therefore, in addition to the strong threat model of this paper about the knowledge of the defender, it is not clear how we compare our method with this paper. In another parallel work, Zhang et al. [4] have also introduced another model extraction detection-based method. However, we still believe that PRADA is the only defense that is comparable to HODA. [2] and [4] are parallel to our work, and we had not enough time to compare HODA with them. We have added a paragraph to the Related Work section of the manuscript to describe these three studies and their limitations.
> >
> > **Q4. Minor inconsistent naming convention, e.g.: CIFAR10 and Cifar10.**
> >
> > **Response**: We have checked the manuscript and corrected typos and abbreviations.
> >
> > **Q5. Why is MobileNet performing the best across the 3 models in Figure 2?**
> >
> > **Response**: We first want to mention that since Figure 2 is not directly relevant to the main purpose of the paper, we have moved that figure to Appendix B and changed the figure to avoid misunderstanding. As mentioned in the paper, calculating the hardness degree of samples does not require the true label of samples. Nevertheless, since we had access to the true labels of CIFAR10 and CIFAR100 test samples, we depict Figure 2 (Figure 6) to give more intuition about the hardness degree of samples. To depict this figure, we first calculate the hardness degree of CIFAR10 and CIFAR100 test samples and then partition them into ten groups based on their hardness degree. Notably, the number of samples in each group is different. Afterward, we calculate what percentage of samples in each group is classified incorrectly. The figure demonstrates that the misclassification rate is increased by increasing the hardness degree of samples. In other words, there is a strong positive correlation between the hardness degree of samples and the misclassification rate.
> >
> > Although the curve of MobileNet is upper than two other classifiers in Figure 2, this does not mean that the MobileNet is the best classifier across the three models. The number of samples in each range of hardness degree is different between models. Given Figure 1, CIFAR10 and CIFAR100 test samples are relatively harder for MobileNet classifiers than two other classifiers. Since models misclassify harder samples with higher probability, and since the number of harder samples is more for the MobileNet classifier, this classifier has the worst performance across models. Figure 6 indicates the percentage of samples in each hardness degree range by a green curve. We have added an example to Appendix B to clarify Figure 6.
> >
> > **Q6. Many transfer learning methods freeze all but the last (classification) layer.**
> >
> > **Response**: Due to the limited time of rebuttal, we cannot train new Caltech256 and CUB200 target classifiers and replicate all experiments on them. It is noteworthy to mention that the accuracy of our Caltech256 (77.2% ours / 78.8% K.Net) and CUB200 (73.7% ours / 76.5% K.Net) target classifiers is comparable to the accuracy of these target classifiers in [5] ([5] trains ResNet34 model in 200 epochs but we train ResNet18 model in 100 epochs).  Generally, we think if the target model uses transfer learning methods that freeze all but the last (classification) layer, the number of easier samples is increased due to the limited update in the target model parameters (decision boundaries).
> >
> > **Q7. In Appendix G, PGD examples are not hard with hardness degree < 70.**
> >
> > **Response**: We corrected it.
> >
> > **References**:
> >
> > [1] S. Kariyappa, et al.. Protecting dnns from theft using an ensemble of diverse models. In ICLR, 2021b.
> >
> > [2] S. Pal, et al.. Stateful detection of model extraction attacks. CoRR, abs/2107.05166, 2021. URL https://arxiv.org/abs/2107.05166.
> >
> > [3] G. Atli, et al.. Extraction of complex dnn models: Real threat or boogeyman? In Engineering Dependable and Secure Machine Learning Systems, pp. 42–57, 2020.
> >
> > [4] Z. Zhang, et al.. Seat: Similarity encoder by adversarial training for detecting model extraction attack queries. In Proceedings of the 14th ACM Workshop AISec, pp. 37–48, 2021.
> >
> > [5] T. Orekondy, et al.. Knockoff nets: Stealing functionality of black-box models. In CVPR, 2019.

---

> > > ### Comment · Reviewer_qQMY · 2021-11-30
> > > **Overhead**
> > >
> > > Thank you for your detailed answers, I appreciate it. Regarding the overheads, HODA still requires additional storage for 11 models and higher GPU time (11 inference passes instead of 1 per sample). On the other hand, I agree that the overall cost of this defense is reasonable.

---

> > > > ### Author Response · Authors · 2021-12-04
> > > > **Overhead**
> > > >
> > > > Dear reviewer qQMY,
> > > >
> > > > We truly appreciate your reply. You are right. HODA requires additional storage for 11 models and higher GPU time. In the revised manuscript, we showed that HODA-5 is also effective against model extraction attacks.
> > > > HODA-5 uses 5 subclassifiers to compute the hardness degree of samples and it requires additional storage for just 5 models.
> > > >
> > > > Best regards.

---

> > > > ### Author Response · Authors · 2021-12-09
> > > > **Thank you.**
> > > >
> > > > Dear reviewer qQMY,
> > > >
> > > > Considering your positive reply, we would really appreciate it if you could please consider raising your score.
> > > >
> > > > Thank you.

---

### Author Response · Authors · 2021-11-28
**A Gentle Reminder**

Dear Reviewers,


We truly appreciate your efforts in reviewing our paper and your constructive comments. Given that the ICLR final discussion deadline is approaching, we really hope to have a further discussion with you to see if our responses address your concerns.  We are happy to address any remaining concerns and eagerly look for your feedback on the revised paper.


Sincerely,

Authors

---

### Decision · Program_Chairs · 2022-01-20

**Decision:**

Reject

**Comment:**

The paper presents a new method for detection of model extraction attacks. It is based on the intuition that typical model extraction attacks involve samples submitted by users that are harder to classify than "benign" samples submitted by users. By introducing the notion of hardness, a metric is developed for identifying malicious users submitting their samples for the purpose of model extraction. While the proposed method is original, it incurs a substantial overhead. Experimental evaluation of the proposed method also has several deficiencies, in particular, in the assessment of its overhead as well as in modeling of benign users.